# A Unified Approach to Reinforcement Learning, Quantal Response Equilibria, and Two-Player Zero-Sum Games

**Samuel Sokota**\*
Carnegie Mellon University
ssokota@andrew.cmu.edu

**Ryan D'Orazio**\*
Mila, Université de Montréal
ryan.dorazio@mila.quebec

**J. Zico Kolter**
Carnegie Mellon University
zkolter@cs.cmu.edu

**Nicolas Loizou**
Johns Hopkins University
nloizou@jhu.edu

**Marc Lanctot**
DeepMind
lanctot@deepmind.com

**Ioannis Mitliagkas**
Mila, Université de Montréal
ioannis@mila.quebec

**Noam Brown**
Meta AI
noambrown@meta.com

**Christian Kroer**
Columbia University
ck2945@columbia.edu

## Abstract

This work studies an algorithm, which we call magnetic mirror descent, that is inspired by mirror descent and the non-Euclidean proximal gradient algorithm. Our contribution is demonstrating the virtues of magnetic mirror descent as both an equilibrium solver and as an approach to reinforcement learning in two-player zero-sum games. These virtues include: 1) Being the first quantal response equilibria solver to achieve linear convergence for extensive-form games with first order feedback; 2) Being the first standard reinforcement learning algorithm to achieve empirically competitive results with CFR in tabular settings; 3) Achieving favorable performance in 3x3 Dark Hex and Phantom Tic-Tac-Toe as a self-play deep reinforcement learning algorithm.

## 1 Introduction

This work studies an algorithm that we call magnetic mirror descent (MMD) in the context of two-player zero-sum games. MMD is an extension of mirror descent (Beck & Teboulle, 2003; Nemirovsky & Yudin, 1983) with proximal regularization and a special case of a non-Euclidean proximal gradient method (Tseng, 2010; Beck, 2017)—both of which have been studied extensively in convex optimization. To facilitate our analysis of MMD, we extend the non-Euclidean proximal gradient method from convex optimization to 2p0s games and variational inequality problems (Facchinei & Pang, 2003) more generally. We then prove a new linear convergence result for the non-Euclidean proximal gradient method in variational inequality problems with composite structure. As a consequence of our general analysis, we attain formal guarantees for MMD by showing that solving for quantal response equilibria (McKelvey & Palfrey, 1995) (i.e., entropy regularized Nash equilibria) in extensive-form games (EFGs) can be modeled as variational inequality problems via the sequence form (Romanovskii, 1962; Von Stengel, 1996; Koller et al., 1996). These guarantees provide the first linear convergence results to quantal response equilibria (QREs) in EFGs for a first order method.

Our empirical contribution investigates MMD as a last iterate (regularized) equilibrium approximation algorithm across a variety of 2p0s benchmarks. We begin by confirming our theory—showing that MMD converges exponentially fast to QREs in both NFGs and EFGs. We also find that, empirically, MMD converges to agent QREs (AQREs) (McKelvey & Palfrey, 1998)—an alternative formulation of QREs for extensive-form games—when applied with action-value feedback. These results lead us to examine MMD as an RL algorithm for approximating Nash equilibria. On this front, we show

---

\*Equal contribution

competitive performance with counterfactual regret minimization (CFR) (Zinkevich et al., 2007). This is the first instance of a standard RL algorithm[1] yielding empirically competitive performance with CFR in tabular benchmarks when applied in self play. Motivated by our tabular results, we examine MMD as a multi-agent deep RL algorithm for 3x3 Abrupt Dark Hex and Phantom Tic-Tac-Toe—encouragingly, we find that MMD is able to successfully minimize an approximation of exploitability. In addition to those listed above, we also provide numerous other experiments in the appendix. In aggregate, we believe that our results suggest that MMD is a unifying approach to reinforcement learning, quantal response equilibria, and two-player zero-sum games.

## 2 BACKGROUND

Sections 2.1 and 3.3 provide a casual treatment of our problem settings and solution concepts and a summary of our algorithm and some of our theoretical results. Sections 2.2 through 3.2 give a more formal and detailed treatment of the same material—these sections are self-contained and safe-to-skip for readers less interested in our theoretical results.

### 2.1 PROBLEM SETTINGS AND SOLUTION CONCEPTS

This work is concerned with 2p0s games—i.e., settings with two players in which the reward for one player is the negation of the reward for the other player.[2] Two-player zero-sum games are often formalized as NFGs, partially observable stochastic games (Hansen et al., 2004) or a perfect-recall EFGs (von Neumann & Morgenstern, 1947). An important idea is that it is possible to convert any EFG into an equivalent NFG. The actions of the equivalent NFG correspond to the deterministic policies of the EFG. The payoffs for a joint action are dictated by the expected returns of the corresponding joint policy in the EFG.

We introduce the solution concepts studied in this work as generalizations of single-agent solution concepts. In single-agent settings, we call these concepts optimal policies and soft-optimal policies. We say a policy is optimal if there does not exist another policy achieving a greater expected return (Sutton & Barto, 2018). In problems with a single decision-point, we say a policy is $\alpha$-soft optimal in the normal sense if it maximizes a weighted combination of its expected action value and its entropy:

$$\pi = \arg\max_{\pi' \in \Delta(\mathbb{A})} \mathbb{E}_{A \sim \pi'} q(A) + \alpha \mathcal{H}(\pi'), \tag{1}$$

where $\pi$ is a policy, $\Delta(\mathbb{A})$ is the action simplex, $q$ is the action-value function, $\alpha$ is the regularization temperature, and $\mathcal{H}$ is Shannon entropy. More generally, we say a policy is $\alpha$-soft optimal in the behavioral sense if it satisfies equation (1) at every decision point.

In 2p0s settings, we refer to the solution concepts used in this work as Nash equilibria and QREs. We say a joint policy is a Nash equilibrium if each player's policy is optimal, conditioned on the other player not changing its policy. In games with a single-decision point, we say a joint policy is a QRE[3] (McKelvey & Palfrey, 1995) if each player's policy is soft optimal in the normal sense, conditioned on the other player not changing its policy. More generally, we say a joint policy is an agent QRE (AQRE) (McKelvey & Palfrey, 1998) if each player's policy is soft optimal in the behavioral sense, subject to the opponent's policy being fixed. Note that AQREs of EFGs do not generally correspond with the QREs of their normal-form equivalents.

Outside of (A)QREs, our results also apply to other regularized solution concepts, such as those having KL regularization toward a non-uniform policy.

### 2.2 NOTATION

We use superscript to denote a particular coordinate of $x = (x^1, \cdots, x^n) \in \mathbb{R}^n$ and subscript to denote time $x_t$. We use the standard inner product denoted as $\langle x, y \rangle = \sum_{i=1}^{n} x^i y^i$. For a given

---

[1]We use "standard RL algorithm" to mean algorithms that would look ordinary to single-agent RL practitioners—excluding, e.g., algorithms that converge in the average iterate or operate over sequence form.

[2]Note that 2p0s games generalize single-agent settings, such as Markov decision processes (Puterman, 2014) and partially observable Markov decision processes (Kaelbling et al., 1998).

[3]Specifically, it is a logit QRE; We omit "logit" as a prefix for brevity.

norm $\|\cdot\|$ on $\mathbb{R}^n$ we define its dual norm $\|y\|_* = \sup_{\|x\|=1}\langle y, x\rangle$. For example, the dual norm to $\|x\|_1 = \sum_{i=1}^{n}|x^i|$ is $\|x\|_\infty = \max_i |x^i|$. We assume all functions $f : \mathbb{R}^n \to (-\infty, +\infty]$ to be closed, with domain of $f$ as $\text{dom} f = \{x : f(x) < +\infty\}$ and corresponding interior $\text{int dom} f$. If $f$ is convex and differentiable, then its minimum $x_* \in \arg\min_{x\in C} f(x)$ over a closed convex set $C$ satisfies $\langle \nabla f(x_*), x - x_*\rangle \geqslant 0$ for any $x \in C$.

We use the Bregman divergence of $\psi$ to generalize the notion of distance. Let $\psi$ be a convex function differentiable over $\text{int dom} \psi$. Then the Bregman divergence with respect to $\psi$ is $B_\psi \colon \text{dom} \psi \times \text{int dom} \psi \to \mathbb{R}$, defined as $B_\psi(x; y) = \psi(x) - \psi(y) - \langle\nabla\psi(y), x - y\rangle$. We say that $f$ is $\mu$-strongly convex over $C$ with respect to $\|\cdot\|$ if $B_f(x;y) \geqslant \frac{\mu}{2}\|x - y\|^2$ for any $x \in C, y \in C \cap \text{int dom} \psi$. Similarly we define relative strong convexity (Lu et al., 2018). We say $g$ is $\mu$-strongly convex relative to $\psi$ over $C$ if $\langle\nabla g(x) - \nabla g(y), x - y\rangle \geqslant \mu\langle\nabla\psi(x) - \nabla\psi(y), x - y\rangle$ or, equivalently, if $B_g(x;y) \geqslant \mu B_\psi(x;y), \forall x, y \in \text{int dom} \psi \cap C$ (Lu et al., 2018). Note both $\psi$ and $B_\psi(\cdot; y)$ are 1-strongly convex relative to $\psi$.

## 2.3 ZERO-SUM GAMES AND QREs

In 2p0s games, the solution of a QRE can be written as the solution to a negative entropy regularized saddle point problem. To model QREs (and more), we consider the regularized min max problem

$$\min_{x\in\mathcal{X}} \max_{y\in\mathcal{Y}} \quad \alpha g_1(x) + f(x, y) - \alpha g_2(y), \tag{2}$$

where $\mathcal{X} \subset \mathbb{R}^n$, $\mathcal{Y} \subset \mathbb{R}^m$ are closed and convex (and possibly unbounded) and $g_1 : \mathbb{R}^n \to \mathbb{R}$, $g_2 : \mathbb{R}^m \to \mathbb{R}$, $f : \mathbb{R}^n \times \mathbb{R}^m \to \mathbb{R}$. Moreover, $g_1$ and $f(\cdot, y)$ are differentiable and convex for every $y$. Similarly $-g_2$, $f(x, \cdot)$ are differentiable and concave for every $x$. A solution $(x_*, y_*)$ to equation 2 is a Nash equilibrium in the regularized game with the following best response conditions along with their equivalent first order optimality conditions

$$x_* \in \arg\min_{x\in\mathcal{X}} \alpha g_1(x) + f(x, y_*) \Leftrightarrow \langle\alpha\nabla g_1(x_*) + \nabla_{x_*}f(x_*, y_*), x - x_*\rangle \geqslant 0 \, \forall x \in \mathcal{X}, \tag{3}$$

$$y_* \in \arg\min_{y\in\mathcal{Y}} \alpha g_2(y) - f(x_*, y) \Leftrightarrow \langle\alpha\nabla g_2(y_*) - \nabla_{y_*}f(x_*, y_*), y - y_*\rangle \geqslant 0 \, \forall y \in \mathcal{Y}. \tag{4}$$

In the context of QREs we have that $\mathcal{X} = \Delta^n, \mathcal{Y} = \Delta^m$ with $f(x, y) = x^\top Ay$ for some payoff matrix $A$, and $g_1, g_2$ are negative entropy. The corresponding best response conditions (3-4) can be written in closed form as $x_* \propto \exp(-Ay_*/\alpha)$, $y_* \propto \exp(A^\top x_*/\alpha)$. Similarly, for EFGs, normal-form QREs take the form of equation 2 (Ling et al., 2018) with $g_1, g_2$ being dilated entropy (Hoda et al., 2010), $f(x, y) = x^\top Ay$ ($A$ being the sequence-from payoff matrix), and $\mathcal{X}, \mathcal{Y}$ the sequence-form strategy spaces of both players.

## 2.4 CONNECTION BETWEEN ZERO-SUM GAMES AND VARIATIONAL INEQUALITIES

More generally, solutions to equation 2 (including QREs) can be written as solutions to variational inequalities (VIs) with specific structure. The equivalent VI formulation stacks both first-order best response conditions (3-4) into one inequality.

**Definition 2.1** (Variational Inequality Problem (VI)). Given $\mathcal{Z} \subseteq \mathbb{R}^n$ and mapping $G : \mathcal{Z} \to \mathbb{R}^n$, the variational inequality problem $\text{VI}(\mathcal{Z}, G)$ is to find $z_* \in \mathcal{Z}$ such that

$$\langle G(z_*), z - z_*\rangle \geqslant 0 \quad \forall z \in \mathcal{Z}. \tag{5}$$

In particular, the optimality conditions (3-4) are equivalent to $\text{VI}(\mathcal{Z}, G)$ where $G = F + \alpha\nabla g$, $\mathcal{Z} = \mathcal{X} \times \mathcal{Y}$ and $g : \mathcal{Z} \to \mathbb{R}, (x, y) \mapsto g_1(x) + g_2(y)$, with corresponding operators $F(z) = [\nabla_x f(x, y), -\nabla_y f(x, y)]^\top$, and $\nabla g = [\nabla_x g_1(x), \nabla_y g_2(y)]^\top$. For more details see Facchinei & Pang (2003)(Section 1.4.2). Note that VIs are more general than min-max problems; they also include fixed-point problems and Nash equilibria in $n$-player general-sum games (Facchinei & Pang, 2003). However, in the case of convex-concave zero-sum games and convex optimization, the problem admits efficient algorithms since the corresponding operator $G$ is *monotone* (Rockafellar, 1970).

**Definition 2.2.** $G$ is said to be strongly monotone if, for $\mu > 0$ and any $z, z'$ where $G$ is defined, $\langle G(z) - G(z'), z - z'\rangle \geqslant \mu\|z - z'\|^2$. G is monotone if this is true for $\mu = 0$.

**Definition 2.3.** $G$ is said to be $L$-smooth with respect to $\|\cdot\|$ if, for any $z, z'$ where $G$ is defined, $\|G(z) - G(z')\|_* \leqslant L\|z - z'\|$.

For EFGs, Ling et al. (2018) showed that the QRE is the solution of a min-max problem of the form equation 2 where $f$ is bilinear and each $g_i$ could be non smooth. Therefore, we can write the problem as a VI with strongly monotone operator $G$ having composite structure, a smooth part coming from $f$ and non-smooth part from the regularization $g_1, g_2$.

**Proposition 2.4.** *Solving a normal-form reduced QRE in a two-player zero-sum EFG is equivalent to solving* $\mathrm{VI}(\mathcal{Z}, F + \alpha \nabla \psi)$ *where* $\mathcal{Z}$ *is the cross-product of the sequence form strategy spaces and* $\psi$ *is the sum of the dilated entropy functions for each player. The function* $\psi$ *is strongly convex with respect to* $\|\cdot\|$. *Furthermore,* $F$ *is monotone and* $\max_{ij} |A_{ij}|$-*smooth (A being the sequence-form payoff matrix) with respect to* $\|\cdot\|$ *and* $F + \alpha \nabla \psi$ *is strongly monotone.*

## 3 ALGORITHMS AND THEORY

In Proposition 2.4, we provided a new perspective to QRE problems that draws connections to VIs with special composite structure. Motivated by this connection, in Section 3.1, we consider an approach to solve such problems via a non-Euclidean proximal gradient method Tseng (2010); Beck (2017) and prove a novel linear convergence result. Thereafter, in Section 3.2, we demonstrate how this general algorithm specializes to MMD and splits into two decentralized simultaneous updates in 2p0s games (one for each player). Finally, in Section 3.3, we discuss specific instances of MMD, give new algorithms for RL and QRE solving, and summarize our linear convergence result for QREs.

### 3.1 CONVERGENCE ANALYSIS

We now present our main algorithm, a non-Euclidean proximal gradient method to solve $\mathrm{VI}(\mathcal{Z}, F + \alpha \nabla g)$. Since $\nabla g$ is possibly not smooth, we incorporate $g$ as a proximal regularization.

**Algorithm 3.1.** *Starting with* $z_1 \in \mathrm{int\,dom}\,\psi \cap \mathcal{Z}$ *at each iteration* $t$ *do*

$$z_{t+1} = \arg\min_{z \in \mathcal{Z}} \eta \left( \langle F(z_t), z \rangle + \alpha g(z) \right) + B_\psi(z; z_t).$$

To ensure that $z_{t+1}$ is well defined, we make the following assumption.

**Assumption 3.2** (Well-defined). Assume $\psi$ is 1-strongly convex with respect to $\|\cdot\|$ over $\mathcal{Z}$ and, for any $\ell$, stepsize $\eta > 0$, $\alpha > 0$, $z_{t+1} = \arg\min_{z \in \mathcal{Z}} \eta \left( \langle \ell, z \rangle + \alpha g(z) \right) + B_\psi(z; z_t) \in \mathrm{int\,dom}\,\psi$.

We also make some assumptions on $F$ and $g$.

**Assumption 3.3.** Let $F$ be monotone and $L$-smooth with respect to $\|\cdot\|$ and $g$ be 1-strongly convex relative to $\psi$ over $\mathcal{Z}$ with $g$ differentiable over $\mathrm{int\,dom}\,\psi$.

These assumptions imply $F + \alpha \nabla g$ is strongly monotone[4] with unique solution $z_*$ (Bauschke et al., 2011). Our result shows that, if $z_* \in \mathrm{int\,dom}\,\psi$[5], then Algorithm 3.1 converges linearly to $z_*$.

**Theorem 3.4.** *Let Assumptions 3.2 and 3.3 hold and assume the unique solution* $z_*$ *to* $\mathrm{VI}(\mathcal{Z}, F + \alpha \nabla g)$ *satisfies* $z_* \in \mathrm{int\,dom}\,\psi$. *Then Algorithm 3.1 converges if* $\eta \leqslant \frac{\alpha}{L^2}$ *and guarantees*

$$B_\psi(z_*; z_{t+1}) \leqslant \left( \frac{1}{1 + \eta \alpha} \right)^t B_\psi(z_*; z_1).$$

Note $\alpha > 0$ is necessary to converge to the solution. If $\alpha = 0$ in the context of solving equation 2, Algorithm 3.1 with $\psi(z) = \frac{1}{2} \|z\|^2$ becomes projected gradient descent ascent, which is known to diverge or cycle for any positive stepsize. However, choosing the strong convexity constants of $g$ and $\psi$ to be 1 is for convenience—the theorem still holds with arbitrary constants, in which case the stepsize condition becomes proportional to the relative strong convexity constant of $g$ (see Corollary D.6 for details).

Due to the generality of VIs, we have the following convex optimization result.

**Corollary 3.5.** *Consider the composite optimization problem* $\min_{z \in \mathcal{Z}} f(z) + \alpha g(z)$. *Then under the same assumptions as Theorem 3.4 with* $F = \nabla f$, *Algorithm 3.1 converges linearly to the solution.*

Note that Corollary 3.5 guarantees linear convergence, which is faster than existing results (Tseng, 2010; Bauschke et al., 2017; Hanzely et al., 2021), due to the additional assumption that $g$ is relatively-strongly convex.

---

[4]This follows because Assumptions (3.2-3.3) imply $g$ is strongly convex and hence $\nabla g$ is strongly monotone.

[5]This assumption is guaranteed in the QRE setting where $g$ is the sum of dilated entropy.

## 3.2 Application of Magnetic Mirror Descent to Two-Player Zero-Sum Games

We define MMD to be Algorithm 3.1 with $g$ taken to be either $\psi$ or $B_\psi(\cdot; z')$ for some $z'$; in both cases the 1-relative strongly convex assumption is satisfied, and $z_{t+1}$ is attracted to either $\min_{z \in \mathcal{Z}} \psi(z)$ or $z'$, which we call the magnet.

**Algorithm 3.6** (Magnetic Mirror Descent (MMD)).

$$z_{t+1} = \underset{z \in \mathcal{Z}}{\arg\min} \, \eta \left( \langle F(z_t), z \rangle + \alpha \psi(z) \right) + B_\psi(z; z_t) \tag{6}$$

*or*

$$z_{t+1} = \underset{z \in \mathcal{Z}}{\arg\min} \, \eta \left( \langle F(z_t), z \rangle + \alpha B_\psi(z; z') \right) + B_\psi(z; z_t). \tag{7}$$

**Remark 3.7.** MMD has the same computational cost as mirror descent since the updates can be equivalently written as $z_{t+1} = \arg\min_{z \in \mathcal{Z}} \langle \ell, z \rangle + \psi(z)$ (e.g. $\ell = (\eta F(z_t) - \nabla \psi(x_t))/(1 + \eta \alpha)$ for equation 6). In fact, Proposition D.7 shows that MMD is equivalent to mirror descent on the regularized loss with a different stepsize.

MMD and, more generally, Algorithm 3.1 can be used to derive a descent-ascent method to solve the zero-sum game equation 2. If $g_1 = \psi_1$ and $g_2 = \psi_2$ are strongly convex over $\mathcal{X}$ and $\mathcal{Y}$, then we can let $\psi(z) = \psi_1(x) + \psi_2(y)$, which makes $\psi$ strongly convex over $\mathcal{Z}$. Then the MMD update rule equation 6 converges to the solution of equation 2 and splits into simultaneous descent-ascent updates:

$$x_{t+1} = \underset{x \in \mathcal{X}}{\arg\min} \, \eta \left( \langle \nabla_{x_t} f(x_t, y_t), x \rangle + \alpha \psi_1(x) \right) + B_{\psi_1}(x; x_t), \tag{8}$$

$$y_{t+1} = \underset{y \in \mathcal{Y}}{\arg\max} \, \eta \left( \langle \nabla_{y_t} f(x_t, y_t), y \rangle - \alpha \psi_2(y) \right) - B_{\psi_2}(y; y_t). \tag{9}$$

## 3.3 Magnet Mirror Descent Summary

MMD's update is parameterized by four objects: a stepsize $\eta$, a regularization temperature $\alpha$, a mirror map $\psi$, and a magnet, which we denote as either $\rho$ or $\zeta$ depending on the $\psi$. The stepsize $\eta$ dictates the extent to which moving away from the current iterate is penalized; the regularization temperature $\alpha$ dictates the extent to which being far away from the magnet (i.e., $\rho$ or $\zeta$) is penalized; the mirror map $\psi$ determines how distance is measured.

If we take $\psi$ to be negative entropy, then, in reinforcement learning language, MMD takes the form

$$\pi_{t+1} = \operatorname{argmax}_\pi \mathbb{E}_{A \sim \pi} q_t(A) - \alpha \mathrm{KL}(\pi, \rho) - \frac{1}{\eta} \mathrm{KL}(\pi, \pi_t), \tag{10}$$

where $\pi_t$ is the current policy, $q_t$ is the Q-value vector for time $t$, and $\rho$ is a magnet policy. For parameterized problems, if $\psi = \frac{1}{2} \| \cdot \|_2^2$, MMD takes the form

$$\theta_{t+1} = \operatorname{argmin}_\theta \langle \nabla_{\theta_t} \mathcal{L}(\theta_t), \theta \rangle + \frac{\alpha}{2} \|\theta - \zeta\|_2^2 + \frac{1}{2\eta} \|\theta - \theta_t\|_2^2, \tag{11}$$

where $\theta_t$ is the current parameter vector, $\mathcal{L}$ is the loss, and $\zeta$ is the magnet.

In settings with discrete actions and unconstrained domains, respectively, these instances of MMD possess close forms, as shown below

$$\pi_{t+1} \propto \left[ \pi_t \rho^{\alpha\eta} e^{\eta q_t} \right]^{\frac{1}{1+\alpha\eta}}, \quad \theta_{t+1} = \left[ \theta_t + \alpha\eta\zeta - \eta \nabla_{\theta_t} \mathcal{L}(\theta_t) \right] \frac{1}{1 + \alpha\eta}. \tag{12}$$

Our main result, Theorem 3.4, and Proposition 2.4 imply that if both players simultaneously update their policies using equation (10) with a uniform magnet in 2p0s NFGs, then their joint policy converges to the $\alpha$-QRE exponentially fast. Similarly, in EFGs, if both players use a type of policy called sequence form with $\psi$ taken to be dilated entropy, then their joint policy converges to the $\alpha$-QRE exponentially fast. Both of these results also hold more generally for equilibria induced by non-uniform magnet policies. MMD can also be a considered as a behavioral-form algorithm in which update rule (10) or (11) is applied at each information state. If $\rho$ is uniform, a fixed point of this instantiation is an $\alpha$-AQRE; more generally, fixed points are regularized equilibria (i.e., fixed points of a regularized best response operator).

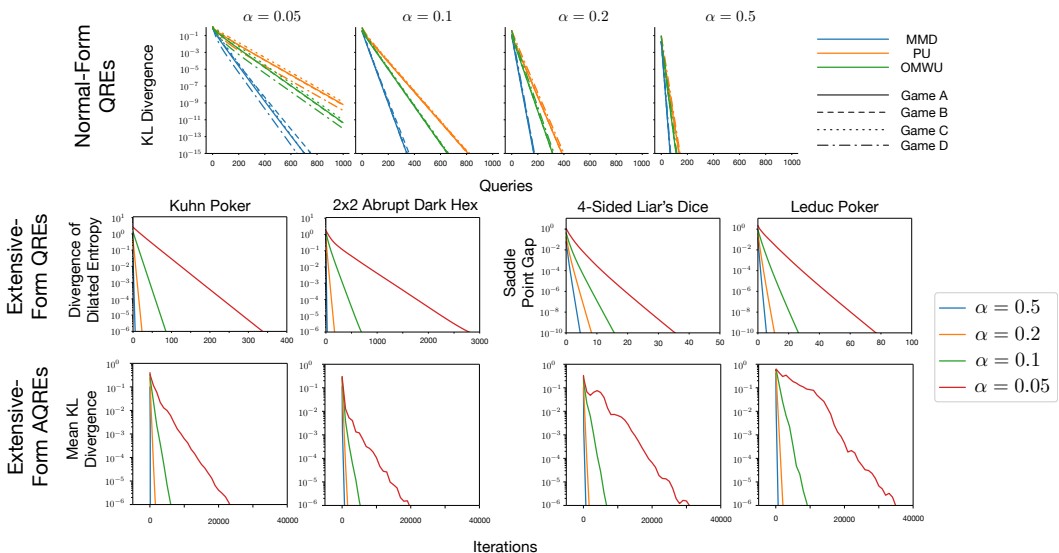

Figure 1: Solving for (A)QREs in various settings.

## 4 EXPERIMENTS

Our main body focuses on highlighting the high level takeaways of our main experiments. Additional discussion of each experiment, as well as additional experiments, are included in the appendix. Code for the sequence form experiments is available at `https://github.com/ryan-dorazio/mmd-dilated`. Code for some of the other experiments is available at `https://github.com/ssokota/mmd`.

**Experimental Domains** For tabular normal-form settings, we used stage games of a 2p0s Markov variant of the game Diplomacy (Paquette et al., 2019). These games have payoff matrices of shape $(50, 50)$, $(35, 43)$, $(50, 50)$, and $(4, 4)$, respectively, and were constructed using an open-source value function (Bakhtin et al., 2021). For tabular extensive-form settings, we used games implemented in OpenSpiel (Lanctot et al., 2019): Kuhn Poker, 2x2 Abrupt Dark Hex, 4-Sided Liar's Dice, and Leduc Poker. These games have 54, 471, 8176, and 9300 non-terminal histories, respectively. For deep multi-agent settings, we used 3x3 Abrupt Dark Hex and Phantom Tic-Tac-Toe, which are also implemented in OpenSpiel.

**Convergence to Quantal Response Equilibria** First, we examine MMD's performance as a QRE solver. We used Ling et al. (2018)'s solver to compute ground truth solutions for NFGs and Gambit (McKelvey, Richard D., McLennan, Andrew M., and Turocy, Theodore L., 2016) to compute ground truth solutions for EFGs. We show the results in Figure 1. We show NFG results in the top row of the figure compared against algorithms introduced by Cen et al. (2021), with each algorithm using the largest stepsize allowed by theory. All three algorithms converge exponentially fast, as is guaranteed by theory. The middle row shows results for QREs on EFG benchmarks. For Kuhn Poker and 2x2 Abrupt Dark Hex, we observe that MMD's divergence converges exponentially fast, as is also guaranteed by theory. For 4-Sided Liar's Dice and Leduc Poker, we found that Gambit had difficulty approximating the QREs, due to the size of the games. Thus, we instead report the saddle point gap (the sum of best response values in the regularized game), for which we observe linear convergence, as is guaranteed by Proposition D.6. The bottom row shows results for AQREs using behavioral form MMD (with $\eta = \alpha/10$) on the same benchmarks, where we also observe convergence (despite a lack of guarantees). For further details, see Sections G.3 for the QRE experiments and Section G.4 for the AQRE experiments.

**Exploitability Experiments** From our AQRE experiments, it immediately follows that it is possible to use behavioral-form MMD with constant stepsize, temperature, and magnet to compute strategies with low exploitabilities.[6] Indeed, we show such results (again with $\eta = \alpha/10$ and a uniform magnet) in the

---

[6]Note that this would also be possible with sequence-form MMD.

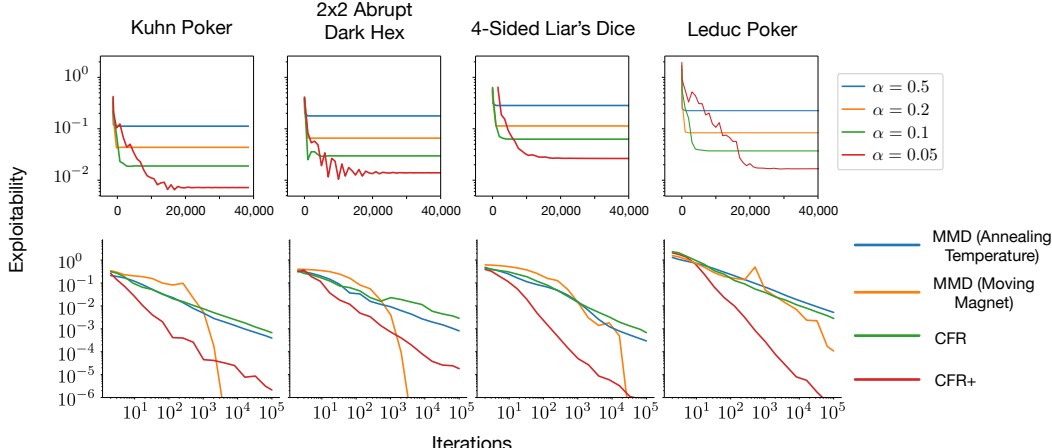

Figure 2: (top) Behavioral-form MMD with constant hyperparameters for various temperatures; (bottom) instances of behavioral-form MMD as Nash equilibria solvers, compared to CFR and CFR+.

top row of Figure 2. A natural follow up question to these experiments is whether MMD can be made into a Nash equilibrium solver by either annealing the amount of regularization over time or by having the magnet trail behind the current iterate. We investigate this question in the bottom row of Figure 2 by comparing i) MMD with an annealed temperature, annealed stepsize, and constant magnet; ii) MMD with a constant temperature, constant stepsize, and moving magnet; iii) CFR (Zinkevich et al., 2007); and iv) CFR+ (Tammelin, 2014). While CFR+ yields the strongest performance, suggesting that it remains the best choice for tabularly solving games, we view the results as very positive. Indeed, not only do both variants of MMD exhibit last-iterate convergent behavior, they also perform competitively with (or better than) CFR. *This is the first instance of a standard RL algorithm yielding results competitive with tabular CFR in classical 2p0s benchmark games.* For further details, see Section H.3 for the annealing temperature experiments and Section H.5 for the moving magnet experiments.

**Deep Multi-Agent Reinforcement Learning** The last experiments in the main body examine MMD as a deep multi-agent RL algorithm using self play. We benchmarked against OpenSpiel's (Lanctot et al., 2019) implementation of NFSP (Heinrich & Silver, 2016) and RLlib's (Liang et al., 2018) implementation of PPO (Schulman et al., 2017). We implemented MMD as a modification of RLlib's (Liang et al., 2018) PPO implementation by changing the adapative forward KL regularization to a reverse KL regularization. For hyperparameters, we tuned $\{(\alpha_t, \eta_t)\}$ for MMD; otherwise, we used default hyperparameters for each algorithm.

As the games are too large to easily compute exact exploitability, we approximate exploitability using a DQN best response, trained for 10 million time steps. The results are shown in the top row of Figure 3. The results include checkpoints after both 1 million and 10 million time steps, as well as bots that select the first legal action (Arbitrary) and that select actions uniformly at random (Random). As expected, both NFSP and MMD yield lower approximate exploitability after 10M steps than they do after 1M steps; on the other hand, PPO does not reliably reduce approximate exploitability over time. In terms of raw value, we find that MMD substantially outperforms the baselines in terms of approximate exploitabilty. We also show results of head-to-head match-ups in the bottom row of Figure 3 for the 10M time step checkpoints. As may be expected given the approximate exploitability results, we find that MMD outperforms our baselines in head-to-head matchups. For further details, see Section J.

**Subject Matter of Additional Experiments** In the appendix, we include 14 additional experiments:

1. Trajectory visualizations for MMD applied as a QRE solver for NFGs (Section D.4);
2. Trajectory visualizations for MMD applied as a saddle point solver (Section D.5)
3. Solving for QREs in NFGs with black box (b.b.) sampling (Section G.2);
4. Solving for QREs in NFGs with b.b. sampling under different gradient estimators (Section G.2);
5. Solving for QREs in Kuhn Poker & 2x2 Abrupt Dark Hex, shown in saddle point gap (Section G.3);
6. Solving for Nash in NFGs with full feedback (Section H.1);
7. Solving for Nash in NFGs with b.b. sampling (Section H.2);
8. Solving for Nash in EFGs with a reach-probability-weighted stepsize (Section H.3);

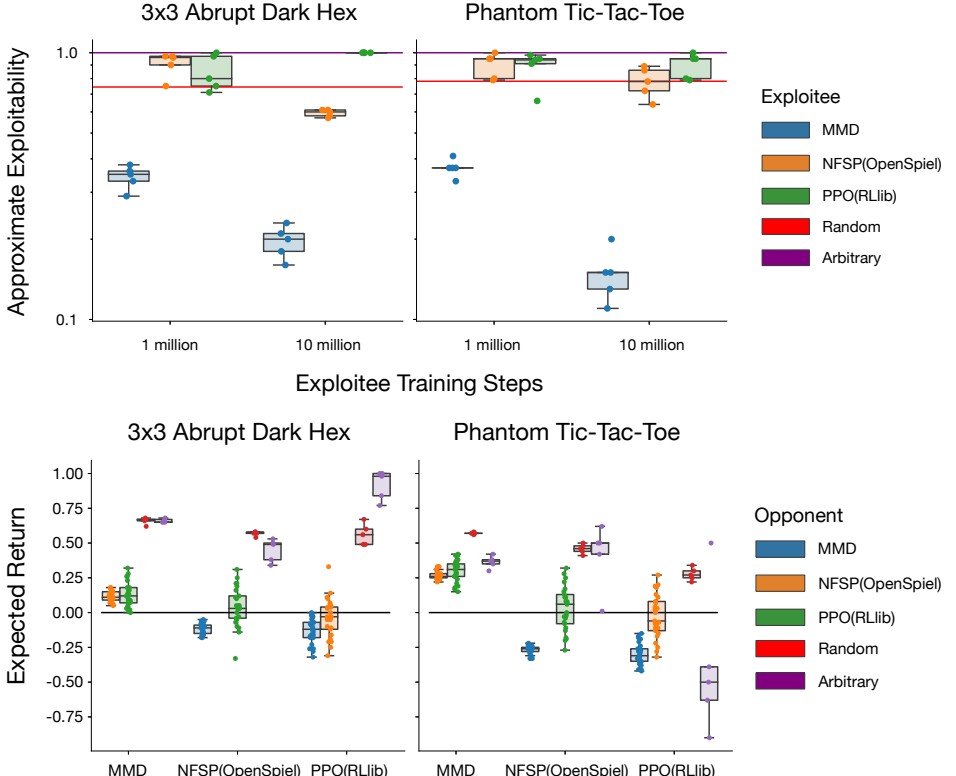

Figure 3: (top) Approximate exploitability experiments; (bottom) head-to-head experiments.

9. Solving for Nash in EFGs with b.b. sampling (Section H.4);
10. Solving for Nash in EFGs using MaxEnt and MiniMaxEnt objectives (Section H.6);
11. Solving for Nash in EFGs with a Euclidean mirror map (Section H.7);
12. Solving for MiniMaxEnt equilibria (Section H.8);
13. Learning in EFGs with constant hyperparameters and a MiniMaxEnt objective (Section H.9).
14. Single-agent deep RL on a small collection Mujoco and Atari games (Section I).

## 5 RELATED WORK

We discuss the main related work below. Additional related work concerning average policy deep reinforcement learning for 2p0s games can be found in Section K.

**Convex Optimization and Variational Inequalities** Like MMD and Algorithm 3.1, the extragradient method (Korpelevich, 1976; Gorbunov et al., 2022) and the optimistic method (Popov, 1980) have also been studied in the context of zero-sum games and variational inequalities more generally. However, in contrast to MMD, these methods require smoothness to guarantee convergence. Outside the context of variational inequalities, analogues of MMD and Algorithm 3.1 have been studied in convex optimization under the non-Euclidean proximal gradient method (Beck, 2017) originally proposed by Tseng (2010). But, in contrast to Theorem 3.4, existing convex optimization results (Beck, 2017; Tseng, 2010; Hanzely et al., 2021; Bauschke et al., 2017) are without linear rates because they do not assume the proximal regularization to be relatively-strongly convex. In addition to convex optimization, the non-Euclidean proximal gradient algorithm has also been studied in online optimization under the name composite mirror descent (Duchi et al., 2010). Duchi et al. (2010) show a $O(\sqrt{t})$ regret bound without strong convexity assumptions on the proximal term. In the case where the proximal term is relatively strongly convex, Duchi et al. (2010) give an improved rate of $O(\log t)$—implying that MMD has average iterate convergence with a rate of $O(\log t/t)$ for bounded problems, like QRE solving.

**Quantal Response Equilibria** Among QRE solvers for NFGs, the PU and OMWPU algorithms from Cen et al. (2021), which also possess linear convergence rates for NFGs, are most similar to MMD. However, both PU and OMWPU require two steps per iteration (because of their similarities to mirror-prox (Nemirovski, 2004) and optimistic mirror descent (Rakhlin & Sridharan, 2013)), and PU requires an extra gradient evaluation. In contrast, our algorithm needs only one simple step per iteration (with the same computation cost as mirror descent) and our analysis applies to various choices of mirror map, meaning our algorithm can be used to compute a larger class of regularized equilibria, rather than only QREs. Among QRE solvers for EFGs, existing algorithms differ from MMD in that they either require second order information (Ling et al., 2018) or are first order methods with average iterate convergence (Farina et al., 2019; Ling et al., 2019). In contrast to these methods, MMD attains linear last-iterate convergence.

**Single-Agent Reinforcement Learning** Considered as a reinforcement learning algorithm, MMD with a negative entropy mirror map and a MaxEnt RL objective coincides with the NE-TRPO algorithm studied in (Shani et al., 2020). MMD with a negative entropy mirror map is also similar to the MD-MPI algorithm proposed by Vieillard et al. (2020) but differs in that MD-MPI includes the negative KL divergence between the current and previous iterate within its Q-values, whereas MMD does not. Considered as a deep reinforcement learning algorithm, MMD with a negative entropy mirror map bears relationships to both KL-PPO (a variant of PPO that served as motivation for the more widely adopted gradient clipping variant) (Schulman et al., 2017) and MDPO (Tomar et al., 2020; Hsu et al., 2020). In short, the negative entropy instantiation of MMD corresponds with KL-PPO with a flipped KL term and with MDPO when there is entropy regularization. We describe these relationships using symbolic expressions in Section L.

**Regularized Follow-the-Regularized-Leader** Another line of work has combined follow-the-regularized-leader with additional regularization, under the names friction follow-the-regularized-leader (F-FoReL) (Pérolat et al., 2021) and piKL (Jacob et al., 2022), in an analogous fashion to how we combine mirror descent with additional regularization.

Similarly to our work, F-FoReL was designed for the purpose of achieving last iterate convergence in 2p0s games. In terms of convergence guarantees, we prove discrete-time linear convergence for NFGs, while Pérolat et al. (2021) give continuous-time linear convergence for EFGs using counterfactual values; neither possesses the desired discrete-time result for EFGs using action values. In terms of ease-of-use, MMD offers the advantage that it is decentralizable, whereas the version of F-FoReL that Pérolat et al. (2021) present is not. In terms of scalability, MMD offers the advantage that it only requires approximating bounded quantities; in contrast, F-FoReL requires estimating an arbitrarily accumulating sum. Lastly, in terms of empirical performance, the tabular results presented in this work for MMD are substantially better than those presented for F-FoReL. For example, F-FoReL's best result in Leduc is an exploitability of about 0.08 after 200,000 iterations—it takes MMD fewer than 1,000 iterations to achieve the same value.

On the other hand, piKL was motivated by improving the prediction accuracy of imitation learning via decision-time planning. We believe the success of piKL in this context suggests that MMD may also perform well in such a setting. While Jacob et al. (2022) also attains convergence to KL-regularized equilibria in NFGs, our results differ in two ways: First, our results only handle the full feedback case, whereas Jacob et al. (2022)'s results allow for stochasticity. Second, our results give linear last-iterate convergence, whereas Jacob et al. (2022) only show $O(\log t/t)$ average-iterate convergence.

## 6 CONCLUSION

In this work, we introduced MMD—an algorithm for reinforcement learning in single-agent settings and 2p0s games, and regularized equilibrium solving. We presented a proof that MMD converges exponentially fast to QREs in EFGs—the first algorithm of its kind to do so. We showed empirically that MMD exhibits desirable properties as a tabular equilibrium solver, as a single-agent deep RL algorithm, and as a multi-agent deep RL algorithm. This is the first instance of an algorithm exhibiting such strong performance across all of these settings simultaneously. We hope that, due to its simplicity, MMD will help open the door to 2p0s games research for RL researchers without game-theoretic backgrounds. We provide directions for future work in Section M.

## 7 ACKNOWLEDGEMENTS

We thank Jeremy Cohen, Chun Kai Ling, Brandon Amos, Paul Muller, Gauthier Gidel, Kilian Fatras, Julien Perolat, Swaminathan Gurumurthy, Gabriele Farina, and Michal Šustr for helpful discussions and feedback. This research was supported by the Bosch Center for Artificial Intelligence, NSERC Discovery grant RGPIN-2019-06512, Samsung, a Canada CIFAR AI Chair, and the Office of Naval Research Young Investigator Program grant N00014-22-1-2530.

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

# Appendices

## Table of Contents

## A  PROBLEM SETTING

In our notation, we use

- $s \in \mathbb{S}$ to notate Markov states,
- $a_i \in \mathbb{A}_i$ to notate actions,
- $o_i \in \mathbb{O}_i$ to notate observations,
- $h_i \in \mathbb{H}_i = \bigcup_t (\mathbb{O}_i \times \mathbb{A}_i)^t \times \mathbb{O}_i$ to denote information states (i.e., decision points).

We use

- $\mathcal{T} \colon \mathbb{S} \times \mathbb{A} \to \Delta(\mathbb{S} \cup \{\perp\})$ to notate the transition function, where $\perp$ notates termination,
- $\mathcal{R}_i \colon \mathbb{S} \times \mathbb{A} \to \mathbb{R}$ to notate a reward function,
- $\mathcal{O}_i \colon \mathbb{S} \times \mathbb{A} \to \mathbb{O}_i$ to notate an observation function.
- $\mathcal{A}_i \colon \mathbb{H}_i \to \mathbb{A}_i$ to notate a legal action function.

We are interested in 2p0s games, in which $i \in \{1, 2\}$ and $\forall s, a, \mathcal{R}_1(s, a) = -\mathcal{R}_2(s, a)$. For convenience, we use $-i$ to notate the player "not $i$". Single-agent settings are captured as a special case in which the second player has a trivial action set $|\mathbb{A}_2| = 1$. Normal-form games are captured as a special case in which there is only one state $s$ and the transition function only supports termination: $\forall a, \mathrm{supp}(\mathcal{T}(s, a)) = \{\perp\}$. (Here, we use $\mathrm{supp}(\mathcal{X})$ to denote the support of a distribution $\mathcal{X}$—i.e., the subset of the domain of $\mathcal{X}$ that is mapped to a value greater than zero: $\{x : \mathcal{X}(x) > 0\}$.)

Each agent's goal is to maximize its expected return

$$\mathbb{E}\left[\sum_t \mathcal{R}_i(S^t, A^t) \mid \pi\right]$$

using its policy $\pi_i$, which dictates a distribution over actions for each information state

$$\pi_i \colon \mathbb{H}_i \to \Delta(\mathbb{A}_i).$$

In game theory literature, these policies are called behavioral form and assume perfect recall.

We notate the expected value for an agent's action $a_i$ at an information state $h_i$ at time $t$ under joint policy $\pi$ as

$$q_\pi(h_i, a_i) = \mathbb{E}\left[\mathcal{R}_i(S, A_{-i}, a_i) + \sum_{t' > t} \mathcal{R}_i(S^{t'}, A^{t'}) \mid \pi, h_i, a_i\right].$$

Here, the first expectation samples the current Markov state $S$ and the current opponent action $A_{-i}$ from the posterior induced by player $i$ reaching information state $h_i$, when each player uses its part of joint policy $\pi$ to determine its actions. The second expectation is over trajectories under the same conditions, with the additional condition that $a_i$ is the agent's action at the current time step.

### A.1  REDUCTION TO NORMAL FORM

Given any game of the above form, we can reduce the game to normal form as follows. Let $\bar{\Pi}_i$ denote the set of deterministic policies—i.e., the set of policies that support exactly one action at a time:

$$\bar{\Pi}_i = \{\pi_i : \forall h_i \, |\mathrm{supp}(\pi_i(h_i))| = 1\}.$$

The action space of the normal-form game is the space of deterministic policies: $\tilde{\mathbb{A}}_i = \bar{\Pi}_i.$[7] The reward function of the normal-form game is dictated by the expected return of the deterministic joint policy.

---

[7]Although the actions $\tilde{\mathbb{A}}_i$ give an equivalent normal-form representation, many of the actions are redundant because actions taken at certain decision points may make other decision points unreachable. The *reduced normal-form* (a.k.a. reduced strategic form) removes duplicate actions by identifying redundant choices at future decision points that are unreachable (Nisan et al., 2007). Hereinafter we consider the reduced normal-form.

**Remark A.1.** Any policy $\pi_i$ can be expressed as a finite mixture over policies in $\bar{\bar{\Pi}}_i$ in a fashion that induces the same distribution over trajectories (against arbitrary, but fixed, opponents). Conversely, any finite mixture over policies in $\bar{\bar{\Pi}}_i$ can be expressed as a policy $\pi_i$ that induces the same distribution over trajectories (against arbitrary, but fixed, opponents).

By the remark above, joint policies in the original game possess counterparts in the normal-form game (and vice versa) achieving identical expected returns. It is in the sense that the normal-form game is equivalent to the original game.

A more detailed exposition on this equivalence can be found in Shoham & Leyton-Brown (2008).

## B  SOLUTION CONCEPTS

Nash equilibria are perhaps the most commonly sought-after solution concept in 2p0s games. A joint policy $\pi_1, \pi_2$ is a Nash equilibrium if neither player can improve its expected return by changing its policy (assuming the other player does not change its policy):

$$\forall i, \pi_i \in \arg\max_{\pi_i'} \mathbb{E}\left[\sum_t \mathcal{R}_i(S^t, A^t) \mid \pi_i', \pi_{-i}\right].$$

Note that, in single-agent settings, this corresponds with the notion of an optimal policy in reinforcement learning.

Another solution concept is a logit quantal response equilibrium (McKelvey & Palfrey, 1995; 1998). As we only deal with logit quantal response equilibria, we generally drop logit and refer to them simply as quantal reponse equilibria. In normal-form games, there are multiple equivalent ways to define a quantal response equilibrium. One way is using entropy regularization. We say a joint policy is an $\alpha$-QRE in a normal-form game if each player maximizes a weighted combination of expected return and policy entropy

$$\forall i, \pi_i \in \arg\max_{\pi_i'} \mathbb{E}\left[\mathcal{R}_i(A) + \alpha\mathcal{H}(\pi_i') \mid \pi_i', \pi_{-i}\right].$$

In a temporally-extended game, we say a joint policy is an $\alpha$-QRE if the equivalent mixture over deterministic joint policies is an $\alpha$-QRE of the equivalent normal-form game.

An alternative way to extend QREs to temporally extended settings is to ask that they satisfy the normal-form QRE condition at each information state:

$$\forall i, \forall h_i, \pi_i(h_i) \in \arg\max_{\pi_i'(h_i)} \mathbb{E}_{A \sim \pi_i'(h_i)}\left[q_i^\pi(h_i, A) + \alpha\mathcal{H}(\pi_i'(h_i))\right]. \tag{13}$$

When a joint policy satisfies this condition, it is called an agent QRE (as it is as if there is a separate agent playing a part of a normal-form QRE at each information state). In single-agent settings, $\alpha$-AQREs correspond with the fixed point of the instantiation of expected SARSA (Sutton & Barto, 2018) in which the policy is a softmax distribution over Q-values with temperature $\alpha$.

The last solution concept that we investigate is called the MiniMaxEnt equilibirum. A joint policy is an $\alpha$-MiniMaxEnt equilibrium if it satisfies condition (13) for MiniMaxEnt $Q$-values

$$q_\pi(h_i, a_i) = \mathbb{E}\left[\mathcal{R}_i(S, A_{-i}, a_i) - \alpha\mathcal{H}(\pi(H_{-i}^t)) + \sum_{t' > t} \mathcal{R}_i(S^{t'}, A^{t'}) + \alpha\mathcal{H}(\pi(H_i^{t'})) - \alpha\mathcal{H}(\pi(H_{-i}^{t'})) \mid \pi, h_i, a_i\right].$$

Alternatively, $\alpha$-MiniMaxEnt equilibria can be defined as the saddlepoint of the $\alpha$-MiniMaxEnt objective

$$\max_{\pi_1}\min_{\pi_2}\mathbb{E}\left[\sum_t \mathcal{R}_1(H^t, A^t) + \alpha\mathcal{H}(\pi_1(H_1^t)) - \alpha\mathcal{H}(\pi_2(H_2^t))\right].$$

While the name MiniMaxEnt is novel to this work, the concept has been studied in recent existing work (Pérolat et al., 2021; Cen et al., 2021).

## C    REDUCED NORMAL-FORM LOGIT-QREs AND MMD

### C.1    SEQUENCE-FORM BACKGROUND

A Nash-equilibrium in a 2p0s extensive-form game can be formulated as a bilinear saddle point problem over the sequence form (Nisan et al., 2007)

$$\min_{x\in\mathcal{X}} \max_{y\in\mathcal{Y}} x^\top Ay,$$

where $\mathcal{X}$ and $\mathcal{Y}$ are the sequence form polytopes, which equivalently can be viewed as treeplexes (Hoda et al., 2010; Kroer et al., 2020). We provide some background on the sequence form in the context of the min player (player 1); the max player follows similarly. Recall all the decision points for player 1 are denoted as $\mathbb{H}_1$ (also known as information states) and the actions available at decision point $h \in \mathbb{H}_1$ are $\mathcal{A}_1(h)$. Recall that a policy (a.k.a behavioral-form strategy) is denoted as $\pi_1$ with $\pi_1(h) \in \Delta(\mathcal{A}_1(h))$ being the policy at decision point $h$. For convenience, let $\pi_1(h, a)$ denote the probability of taking action $a \in \mathcal{A}_1(h)$ at decision point $h$. Next we denote $p(h)$ as the parent sequence to reach decision point $h$—that is, the unique previous decision point and action taken by the player before reaching $h$. Note that this parent is unique due to perfect recall and that it is possible for many decision points to share the same parent. Then we can construct the sequence form from the top down, where the $(h, a)$ sequence of $x \in \mathcal{X}$ is given by

$$x^{(h,a)} = x^{p(h)} \pi(h, a).$$

For convenience, the root sequence $\varnothing$ is defined to be the parent of all initial decision points of the game and is set to the constant $x^\varnothing = 1$. We denote $x^h$ as the slice of $x = (x^{(h,a)})_{h\in\mathbb{H}_1, a\in\mathcal{A}(h)}$ corresponding to decision point $h$. Note we have the following relationship

$$\pi(h) = x^h / x^{p(h)}.$$

Because $x^{(h,a)}$ corresponds to the probability of player 1 choosing *all* actions along the sequence until reaching $(h, a)$, we get that $x^\top Ay$ is the expected payoff for player 2 given a pair of sequence-form strategies $x, y$. Thus the sequence form allows us to get a bilinear objective.

Given the bilinear structure of the sequence-form problem, we convert the problem into a VI using first-order optimality conditions. In order to apply MMD (or other first-order methods), we need a good choice of a mirror map for $\mathcal{X}$ and $\mathcal{Y}$. A such choice is the class of *dilated distance generating functions* (Hoda et al., 2010):

$$\psi(x) = \sum_{h\in\mathbb{H}_1} \beta_h x^{p(h)} \psi_h \left( \frac{x^h}{x^{p(h)}} \right) \tag{14}$$

$$= \sum_{h\in\mathbb{H}_1} \beta_h x^{p(h)} \psi_h(\pi(h)), \tag{15}$$

where $(\beta_h)_{h\in\mathbb{H}_1} > 0$ are per-decision-point weights and $\psi_h$ is a distance-generating function for the simplex associated to $h$. If $\psi_h$ is taken to be the negative entropy then we say $\psi$ is the dilated entropy function. In the normal-form setting the dilated entropy is simply the standard negative entropy.

Recently it was shown that an $\alpha$-QRE (for the reduced normal form) is the solution to the following saddle point problem over the sequence form (Ling et al., 2018),

$$\min_{x\in\mathcal{X}} \max_{y\in\mathcal{Y}} \alpha\psi_1(x) + x^\top Ay - \alpha\psi_2(y), \tag{16}$$

where $\psi_1, \psi_2$ are dilated entropy functions with weights $\beta_h = 1$. Note that we have the normal form $\alpha$-QRE as a special case of equation 16.

### C.2    PROOF FOR PROPOSITION 2.4

**Proposition 2.4.** *Solving a normal-form reduced QRE in a two-player zero-sum EFG is equivalent to solving* $\mathrm{VI}(\mathcal{Z}, F + \alpha\nabla\psi)$ *where $\mathcal{Z}$ is the cross-product of the sequence form strategy spaces and $\psi$ is the sum of the dilated entropy functions for each player. The function $\psi$ is strongly convex with respect to $\|\cdot\|$. Furthermore, $F$ is monotone and $\max_{ij} |A_{ij}|$-smooth (A being the sequence-form payoff matrix) with respect to $\|\cdot\|$ and $F + \alpha\nabla\psi$ is strongly monotone.*

*Proof.* The problem of finding a reduced normal-form logit QRE is equivalent to solving the saddle-point problem stated in equation 16 (Ling et al., 2018). Therefore, due to the convexity of $\psi_1$ and $\psi_2$ and the discussion from Section 2.3, we have that the solution to equation 16 is equivalent to the solution of $\mathrm{VI}(\mathcal{Z}, F + \nabla\psi)$ where,

$$F(z) = \begin{bmatrix} Ay \\ -A^\top x \end{bmatrix}, \quad \nabla\psi(z) = \begin{bmatrix} \nabla_x\psi_1(x) \\ \nabla_y\psi_2(y) \end{bmatrix}. \tag{17}$$

From Hoda et al. (2010) we know there exists constants $\mu_1$, $\mu_2$ such that $\psi_1$ is $\mu_1$-strongly convex over $\mathcal{X}$ with respect to $\|\cdot\|_1$ and $\psi_2$ is $\mu_2$-strongly convex over $\mathcal{Y}$ with respect to $\|\cdot\|_1$ (Hoda et al. (2010) do not show bounds on these constants, but we only need them to exist). Therefore, $\psi$ is also strongly convex over $\mathcal{Z}$ with constant $\min\{\mu_1, \mu_2\}$ with respect to $\|z\| = \sqrt{\|x\|_1^2 + \|y\|_1^2}$ since, for $z = (x, y)$, and $z' = (x', y')$ we have

$$\langle \nabla\psi(z) - \nabla\psi(z'), z - z' \rangle = \langle \nabla\psi_1(x) - \nabla\psi_1(x'), x - x' \rangle + \langle \nabla\psi_2(y) - \nabla\psi_2(y'), y - y' \rangle$$
$$\geqslant \mu_1\|x - x'\|_1^2 + \mu_2\|y - y'\|_1^2$$
$$\geqslant \min\{\mu_1, \mu_2\}\left(\|x - x'\|_1^2 + \|y - y'\|_1^2\right)$$
$$= \min\{\mu_1, \mu_2\}\|z - z'\|^2.$$

Following Theorem 3.4, it is useful to characterize the smoothness of $F$ under the same norm for which $\psi$ is strongly-convex. First, notice that for any matrix $A$ we have that $\|Ax - Ay\|_\infty \leqslant \max_{ij}|A_{ij}|\|x - y\|_1$ (see for example Bubeck et al. (2015)[Section 5.2.4]). Therefore altogether we have

$$\|F(z) - F(z')\|_*^2 = \|Ay - Ay'\|_\infty^2 + \|A^\top x - A^\top x'\|_\infty^2$$
$$\leqslant \max_{ij}|A_{ij}|^2\left(\|y - y'\|_1^2 + \|x - x'\|_1^2\right)$$
$$= \max_{ij}|A_{ij}|^2\|z - z'\|^2,$$

showing that $F$ is $L = \max_{ij}|A_{ij}|$-smooth with respect to $\|\cdot\|$. The strong-monotonicity of $F + \nabla\psi$ follows since $F$ is monotone and $\nabla\psi$ is strongly monotone since $\psi$ is strongly convex. $\square$

Note that in general the Hessian can have unbounded entries (Kroer et al., 2020) meaning that $\nabla\psi$ cannot be $L$-smooth (Beck, 2017). Our MMD algorithm which handles $\psi$ in closed form allows us to sidestep this issue. We also have that the dual norm of $\|\cdot\|$ is simply $\|z\|_* = \sqrt{\|x\|_\infty^2 + \|y\|_\infty^2}$ (Bubeck et al., 2015; Nemirovski, 2004).

## C.3 MMD for Finding Reduced Normal-Form QREs over the Sequence-Form

From Proposition 2.4 and Corollary D.6 we have that the MMD descent-ascent updates (8-9) with $\psi_1, \psi_2$, taken to be dilated entropy with $\eta \leqslant \alpha/\max_{ij}|A_{ij}|^2$ converges linearly to the solution of equation 16. The updates, as mentioned by Remark 3.7, can be computed in closed-form as a one-line change to mirror descent with dilated-entropy (Kroer et al., 2020). Indeed, setting $g_t = Ay_t$ (the gradient for the min player), we have that the update for the min player can be written as follows

$$x_{t+1}\arg\min_{x\in\mathcal{X}} \langle \eta g_t, x \rangle + \eta\alpha\psi(x) + B_\psi(x; x_t)$$
$$= \arg\min_{x\in\mathcal{X}} \langle \eta g_t - \nabla\psi(x_t), x \rangle + (\eta\alpha + 1)\psi(x)$$
$$= \arg\min_{x\in\mathcal{X}} \langle \frac{\eta g_t - \nabla\psi(x_t)}{(1 + \eta\alpha)}, x \rangle + \psi(x)$$
$$= \arg\min_{x\in\mathcal{X}} \sum_{h\in\mathbb{H}_1} \langle \frac{\eta g_t^h - \nabla\psi(x_t)^h}{(1 + \eta\alpha)}, x^h \rangle + x^{p(h)}\psi_h(x^h/x^{p(h)})$$
$$= \arg\min_{x\in\mathcal{X}} \sum_{h\in\mathbb{H}_1} x^{p(h)}\left(\langle \frac{\eta g_t^h - \nabla\psi(x_t)^h}{(1 + \eta\alpha)}, \pi(h) \rangle + \psi_h(\pi(h))\right).$$

Updates can be computed in closed-form starting from decision points $h$ without any children and progressing upwards in the game tree.

# D  PROOFS

## D.1  SUPPORTING LEMMAS AND PROPOSITIONS

**Proposition D.1** ((Bauschke et al., 2003)Proposition 2.3). *Let $\{x, y\} \subset \operatorname{dom} \psi$ and $\{u, v\} \subset \operatorname{int} \operatorname{dom} \psi$. Then*

1. $B_\psi(u; v) + B_\psi(v; u) = \langle \nabla \psi(u) - \nabla \psi(v), u - v \rangle$

2. $B_\psi(x; u) = B_\psi(x; v) + B_\psi(v; u) + \langle \nabla \psi(v) - \nabla \psi(u), x - v \rangle$

3. $B_\psi(x; v) + B_\psi(y; u) = B_\psi(x; u) + B_\psi(y; v) + \langle \nabla \psi(u) - \nabla \psi(v), x - y \rangle.$

The following result is also known as the non-Euclidean prox theorem (Beck, 2017)[Theorem 9.12] or the three-point property(Tseng, 2008).

**Proposition D.2.** *Assume $\mathcal{Z}$ closed convex and both $f$ and $\psi$ are differentiable at $\bar{z}$ (defined below). Then the following statements are equivalent*

1. $\bar{z} = \arg\min_{z \in \mathcal{Z}} \eta \langle g, z \rangle + f(z) + B_\psi(z; y)$

2. $\forall z \in \mathcal{Z} \quad \langle \eta g + \nabla f(\bar{z}), \bar{z} - z \rangle \leqslant B_\psi(z; y) - B_\psi(z; \bar{z}) - B_\psi(\bar{z}, y)$

*Proof.*

$$\bar{z} = \arg\min_{z \in \mathcal{X}} \eta \langle g, z \rangle + f(z) + B_\psi(z; y) \Leftrightarrow \langle \nabla \psi(\bar{z}) + \eta g + \nabla f(\bar{z}) - \nabla \psi(y), z - \bar{z} \rangle \geqslant 0 \quad \forall z \in \mathcal{Z}$$

$$\Leftrightarrow \langle \nabla \psi(y) - \nabla \psi(\bar{z}) - \eta g - \nabla f(\bar{z}), z - \bar{z} \rangle \leqslant 0 \quad \forall z \in \mathcal{Z}$$

$$\Leftrightarrow \langle \eta g + \nabla f(\bar{z}), \bar{z} - z \rangle \leqslant \langle \nabla \psi(\bar{z}) - \nabla \psi(y), z - \bar{z} \rangle \quad \forall z \in \mathcal{Z}$$

$$\Leftrightarrow \langle \eta g + \nabla f(\bar{z}), \bar{z} - z \rangle \leqslant B_\psi(z; y) - B_\psi(z; \bar{z}) - B_\psi(\bar{z}; y) \, \forall z \in \mathcal{Z}.$$

The first equivalence follows by the first-order optimality condition and the last one by Proposition D.1. $\qquad\square$

**Lemma D.3.** *One step of Algorithm 3.1, under the assumptions of Theorem 3.4, guarantees that, for all $z \in \mathcal{Z}$,*

$$B_\psi(z; z_{t+1}) \leqslant \tag{18}$$
$$B_\psi(z; z_t) - B_\psi(z_{t+1}; z_t) + \langle \eta F(z_t) + \eta \alpha \nabla g(z_{t+1}), z - z_{t+1} \rangle. \tag{19}$$

*Proof.* Immediate from Proposition D.2. $\qquad\square$

**Lemma D.4.** *Under the same assumptions as Theorem 3.4, let $z_*$ be the solution to $\mathrm{VI}(\mathcal{Z}, F + \alpha \nabla g)$. Then, for any $z \in \mathcal{Z} \cap \operatorname{int} \operatorname{dom} \psi$, the following inequality holds*

$$\langle \eta F(z) + \eta \alpha \nabla g(z), z_* - z \rangle \leqslant -\eta \alpha \left( B_\psi(z; z_*) + B_\psi(z_*; z) \right).$$

*Proof.*

$$\langle \eta F(z) + \eta \alpha \nabla g(z), z_* - z \rangle = \langle \eta F(z) + \eta \alpha \nabla g(z) - \eta F(z_*) - \eta \alpha \nabla g(z_*), z_* - z \rangle$$
$$+ \langle \eta F(z_*) + \eta \alpha \nabla g(z_*), z_* - z \rangle$$
$$= \underbrace{\langle \eta F(z) - \eta F(z_*), z_* - z \rangle}_{\leqslant 0} + \eta \alpha \langle \nabla g(z) - \nabla g(z_*), z_* - z \rangle$$
$$+ \underbrace{\langle \eta F(z_*) + \eta \alpha \nabla g(z_*), z_* - z \rangle}_{\leqslant 0}$$
$$\leqslant \eta \alpha \langle \nabla g(z) - \nabla g(z_*), z_* - z \rangle$$
$$= -\eta \alpha \langle \nabla g(z) - \nabla g(z_*), z - z_* \rangle$$
$$\leqslant -\eta \alpha \langle \nabla \psi(z) - \nabla \psi(z_*), z - z_* \rangle$$
$$= -\eta \alpha \left( B_\psi(z; z_*) + B_\psi(z_*; z) \right)$$

Note that $\nabla g(z), \nabla g(z_*), \nabla \psi(z), \nabla \psi(z_*)$ are all well-defined because $z_* \in \operatorname{int} \operatorname{dom} \psi$ and Assumptions (3.2-3.3). The first inequality follows since $F$ is monotone and $z_* \in \operatorname{sol} \operatorname{VI}(\mathcal{Z}, F + \alpha \nabla g)$. The second inequality follows since $g$ is 1-strongly convex relative to $\psi$ and the last equality by Proposition D.1. $\qquad \square$

### D.2 PROOF OF THEOREM 3.4

**Theorem 3.4.** *Let Assumptions 3.2 and 3.3 hold and assume the unique solution $z_*$ to $\operatorname{VI}(\mathcal{Z}, F + \alpha \nabla g)$ satisfies $z_* \in \operatorname{int} \operatorname{dom} \psi$. Then Algorithm 3.1 converges if $\eta \leqslant \frac{\alpha}{L^2}$ and guarantees*

$$B_\psi(z_*; z_{t+1}) \leqslant \left(\frac{1}{1+\eta\alpha}\right)^t B_\psi(z_*; z_1).$$

*Proof.*

$$
\begin{aligned}
B_\psi(z_*; z_{t+1}) &\leqslant B_\psi(z_*; z_t) - B_\psi(z_{t+1}; z_t) + \langle \eta F(z_t) + \eta\alpha\nabla g(z_{t+1}), z_* - z_{t+1} \rangle \\
&= B_\psi(z_*; z_t) - B_\psi(z_{t+1}; z_t) + \langle \eta F(z_t) - \eta F(z_{t+1}), z_* - z_{t+1} \rangle + \langle \eta F(z_{t+1}) + \eta\alpha\nabla g(z_{t+1}), z_* - z_{t+1} \rangle \\
&\leqslant B_\psi(z_*; z_t) - B_\psi(z_{t+1}; z_t) + \langle \eta F(z_t) - \eta F(z_{t+1}), z_* - z_{t+1} \rangle - \eta\alpha \left( B_\psi(z_{t+1}; z_*) + B_\psi(z_*; z_{t+1}) \right) \\
&\leqslant B_\psi(z_*; z_t) - B_\psi(z_{t+1}; z_t) + \eta L \|z_t - z_{t+1}\| \|z_* - z_{t+1}\| - \eta\alpha \left( B_\psi(z_{t+1}; z_*) + B_\psi(z_*; z_{t+1}) \right) \\
&\leqslant B_\psi(z_*; z_t) - B_\psi(z_{t+1}; z_t) + \frac{1}{2}\|z_t - z_{t+1}\|^2 + \frac{\eta^2 L^2}{2}\|z_* - z_{t+1}\|^2 - \eta\alpha \left( B_\psi(z_{t+1}; z_*) + B_\psi(z_*; z_{t+1}) \right) \\
&\leqslant B_\psi(z_*; z_t) + \eta^2 L^2 B_\psi(z_{t+1}; z_*) - \eta\alpha \left( B_\psi(z_{t+1}; z_*) + B_\psi(z_*; z_{t+1}) \right) \\
&\overset{\eta^2 L^2 \leqslant \eta\alpha}{\leqslant} B_\psi(z_*; z_t) - \eta\alpha B_\psi(z_*; z_{t+1}).
\end{aligned}
$$

The first inequality follows from Lemma D.3 and the second inequality from Lemma D.4; the third inequality by the generalized Cauchy-Schwarz inequality and the smoothness of $F$; the fourth inequality by elementary inequality $ab \leqslant \frac{\rho a^2}{2} + \frac{b^2}{2\rho} \quad \forall \rho > 0$; and the fifth inequality by the strong convexity of $\psi$ since $\frac{1}{2}\|x - y\|^2 \leqslant B_\psi(x; y)$. Therefore altogether we have

$$B_\psi(z_*; z_{t+1}) \leqslant \frac{B_\psi(z_*; z_t)}{1+\eta\alpha}.$$

Iterating the inequality yields the result. $\qquad \square$

**Corollary D.6.** *Under the same assumptions as Theorem 3.4, if $g$ is $\mu$-strongly convex relative to $\psi$ and $\psi$ is $\mu_\psi$ strongly convex, then if $\eta \leqslant \frac{\alpha\mu}{L^2}$, Algorithm 3.1 guarantees*

$$B_\psi(z_*; z_{t+1}) \leqslant \left(\frac{1}{1+\eta\mu\alpha}\right)^t B_\psi(z_*; z_1).$$

*Proof.* Observe that $\bar{\psi} = \frac{\psi}{\mu_\psi}$ is 1-strongly convex and $\bar{g} = \frac{g}{\mu\mu_\psi}$ is 1-strongly convex relative to $\bar{\psi}$,

$$\langle \nabla \bar{g}(x) - \nabla \bar{g}(y), x - y \rangle = \frac{1}{\mu\mu_\psi}\langle \nabla g(x) - \nabla g(y), x - y \rangle \tag{20}$$

$$\geqslant \frac{1}{\mu_\psi}\langle \nabla \psi(x) - \nabla \psi(y), x - y \rangle \tag{21}$$

$$= \langle \nabla \bar{\psi}(x) - \nabla \bar{\psi}(y), x - y \rangle. \tag{22}$$

Rewriting the update of Algorithm 3.1 in terms of $\bar{g}$ and $\bar{\psi}$ gives

$$\arg\min_{z\in\mathcal{Z}} \eta\left(\langle F(z_t), z\rangle + \alpha g(z)\right) + B_\psi(z; z_t)$$

$$\Leftrightarrow \arg\min_{z\in\mathcal{Z}} \eta\left(\langle F(z_t), z\rangle + \frac{\alpha\mu\mu_\psi}{\mu\mu_\psi}g(z)\right) + \frac{\mu_\psi}{\mu_\psi}B_\psi(z; z_t)$$

$$\Leftrightarrow \arg\min_{z\in\mathcal{Z}} \frac{\eta}{\mu_\psi}\left(\langle F(z_t), z\rangle + \frac{\alpha\mu\mu_\psi}{\mu\mu_\psi}g(z)\right) + \frac{1}{\mu_\psi}B_\psi(z; z_t)$$

$$\Leftrightarrow \arg\min_{z\in\mathcal{Z}} \bar{\eta}\left(\langle F(z_t), z\rangle + \alpha\mu\mu_\psi\bar{g}(z)\right) + B_{\bar{\psi}}(z; z_t)$$

$$\Leftrightarrow \arg\min_{z\in\mathcal{Z}} \bar{\eta}\left(\langle F(z_t), z\rangle + \bar{\alpha}\bar{g}(z)\right) + B_{\bar{\psi}}(z; z_t).$$

The result follows from Theorem 3.4 with stepsize $\bar{\eta} = \frac{\eta}{\mu_\psi}$ and $\bar{\alpha} = \mu\mu_\psi\alpha$.

$\square$

## D.3    EQUIVALENCE BETWEEN MMD AND MD

In this section we show that MMD is equivalent to mirror descent (MD) with a different stepsize when an extra regularized loss is added. In the game context this implies that MMD can be implemented as mirror descent ascent on the regularized game with a particular stepsize.

**Proposition D.7.** *Magnetic mirror descent updates (6, 7) are equivalent to the following updates respectively (where the second update uses a magnet $z'$):*

$$z_{t+1} = \arg\min_{z\in\mathcal{Z}} \bar{\eta}\langle F(z_t) + \alpha\nabla\psi(z_t), z\rangle + B_\psi(z; z_t), \tag{23}$$

$$z_{t+1} = \arg\min_{z\in\mathcal{Z}} \bar{\eta}\langle F(z_t) + \alpha\nabla_{z_t}B_\psi(z_t; z'), z\rangle + B_\psi(z; z_t), \tag{24}$$

*with stepsize $\bar{\eta} = \frac{\eta}{1+\eta\alpha}$.*

*Proof.* We begin by proving the first equivalence, between equation (6) and (23):

$$z_{t+1} = \arg\min_{z\in\mathcal{Z}} \eta\left(\langle F(z_t), z\rangle + \alpha\psi(z)\right) + B_\psi(z; z_t)$$

$$\Leftrightarrow z_{t+1} = \arg\min_{z\in\mathcal{Z}} \langle\eta F(z_t) - \nabla\psi(z_t), z\rangle + (1+\eta\alpha)\psi(z)$$

$$\Leftrightarrow z_{t+1} = \arg\min_{z\in\mathcal{Z}} \langle\frac{\eta F(z_t) - \nabla\psi(z_t)}{1+\eta\alpha}, z\rangle + \psi(z)$$

$$\Leftrightarrow z_{t+1} = \arg\min_{z\in\mathcal{Z}} \langle\frac{\eta F(z_t) + \eta\alpha\nabla\psi(z_t)}{1+\eta\alpha} - \nabla\psi(z_t), z\rangle + \psi(z)$$

$$\Leftrightarrow z_{t+1} = \arg\min_{z\in\mathcal{Z}} \langle\bar{\eta}\left(F(z_t) + \alpha\nabla\psi(z_t)\right) - \nabla\psi(z_t), z\rangle + \psi(z)$$

$$\Leftrightarrow z_{t+1} = \arg\min_{z\in\mathcal{Z}} \bar{\eta}\langle F(z_t) + \alpha\nabla\psi(z_t), z\rangle + B_\psi(z; z_t).$$

The second equivalence follows from similar steps:

$$z_{t+1} = \arg\min_{z\in\mathcal{Z}} \eta\left(\langle F(z_t), z\rangle + \alpha B_\psi(z; z')\right) + B_\psi(z; z_t)$$

$$\Leftrightarrow z_{t+1} = \arg\min_{z\in\mathcal{Z}} \langle\eta F(z_t) - \eta\alpha\nabla\psi(z') - \nabla\psi(z_t), z\rangle + (1+\eta\alpha)\psi(z)$$

$$\Leftrightarrow z_{t+1} = \arg\min_{z\in\mathcal{Z}} \langle\frac{\eta F(z_t) - \eta\alpha\nabla\psi(z') - \nabla\psi(z_t)}{1+\eta\alpha}, z\rangle + \psi(z)$$

$$\Leftrightarrow z_{t+1} = \arg\min_{z\in\mathcal{Z}} \langle\frac{\eta F(z_t) - \eta\alpha\nabla\psi(z') + \eta\alpha\nabla\psi(z_t)}{1+\eta\alpha} - \nabla\psi(z_t), z\rangle + \psi(z)$$

$$\Leftrightarrow z_{t+1} = \arg\min_{z\in\mathcal{Z}} \langle\bar{\eta}\left(F(z_t) + \alpha\nabla\psi(z_t) - \alpha\nabla\psi(z')\right) - \nabla\psi(z_t), z\rangle + \psi(z)$$

$$\Leftrightarrow z_{t+1} = \arg\min_{z\in\mathcal{Z}} \langle\bar{\eta}\left(F(z_t) + \alpha\nabla_{z_t}B_\psi(z_t; z')\right) - \nabla\psi(z_t), z\rangle + \psi(z)$$

$$\Leftrightarrow z_{t+1} = \arg\min_{z\in\mathcal{Z}} \bar{\eta}\langle F(z_t) + \alpha\nabla_{z_t}B_\psi(z_t; z'), z\rangle + B_\psi(z; z_t)..$$

□

### D.4 NEGATIVE ENTROPY MMD EXAMPLE

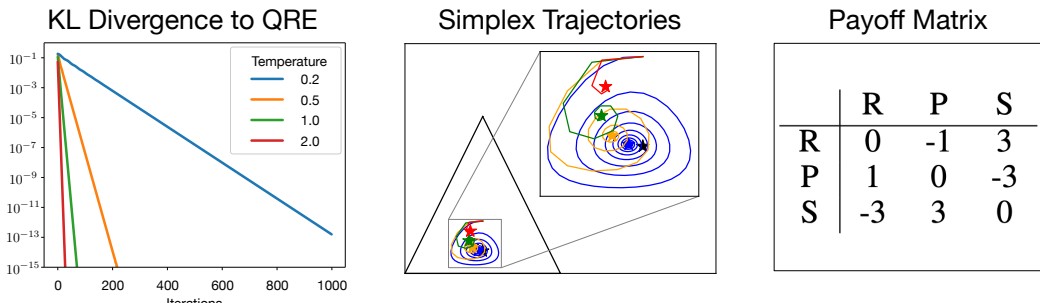

Figure 4: A visualization of convergence in perturbed RPS.

An example showing the simplex trajectories of MMD applied to a small NFG is shown in Figure 4.

### D.5 EUCLIDEAN MMD EXAMPLE

We discuss the Euclidean case for update equation 6 (update 7 is similar). In the Eucldiean case were $\psi = \frac{1}{2}\|\cdot\|_2^2$ we have that update equation 6 reduces to

$$z_{t+1} = \Pi_{\mathcal{Z}}\left(\frac{z_t - \eta F(z_t)}{1 + \eta\alpha}\right),$$

where $\Pi_{\mathcal{Z}}$ denotes the Euclidean projection onto $\mathcal{Z}$. In the context of solving min max problems where $\psi = \psi_1 + \psi_2$, the sum of $\frac{1}{2}\|\cdot\|_2^2$ then the descent-ascent updates of (8,9) become

$$x_{t+1} = \Pi_{\mathcal{X}}\left(\frac{x_t - \eta\nabla_{x_t}f(x_t, y_t)}{1 + \eta\alpha}\right),$$
$$y_{t+1} = \Pi_{\mathcal{Y}}\left(\frac{y_t + \eta\nabla_{y_t}f(x_t, y_t)}{1 + \eta\alpha}\right).$$

Note that our results don't require bounded constraints—in the unconstrained setting there would be no projection step. By Theorem 3.4 the above iterations converge linearly to the solution of

$$\min_{x\in\mathcal{X}}\max_{y\in\mathcal{Y}} \quad \frac{\alpha}{2}\|x\|_2^2 + f(x, y) - \frac{\alpha}{2}\|y\|_2^2,$$

provided $f$ is smooth in the sense that $F(x, y) = [\nabla_x f(x, y), -\nabla_y f(x, y)]^\top$ is smooth.

For example, in the 1-D case we have that the following unconstrained saddle point problem can be solved with Euclidean MMD:

$$\min_{x\in\mathbb{R}}\max_{y\in\mathbb{R}} \frac{\alpha}{2}x^2 + (x - a)(y - b) - \frac{\alpha}{2}y^2.$$

Where $a, b$ are constants. In this case $F(x, y) = [y - b, -(x - a)]^\top$, which is 1-smooth. Therefore, with stepsize $\eta = \alpha$ the following update rule converges linearly to the solution:

$$x_{t+1} = \frac{x_t - \alpha(y_t - b)}{1 + \alpha^2}, \quad y_{t+1} = \frac{y_t + \alpha(x_t - a)}{1 + \alpha^2}.$$

See Figure 5 for a visualization with $a = b = 1$.

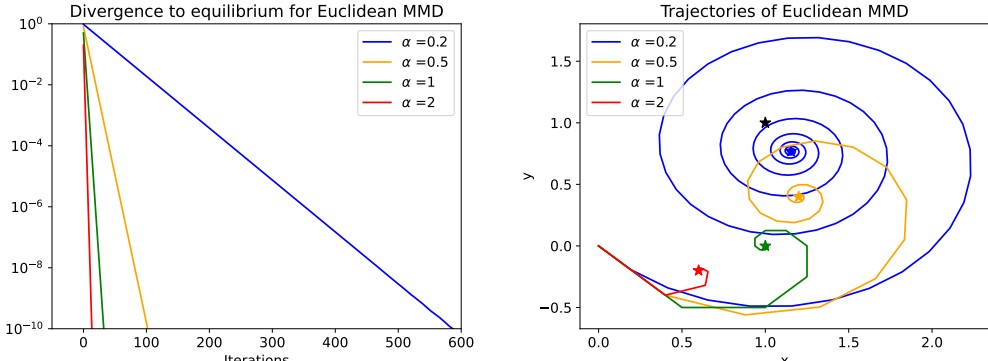

Figure 5: Convergence of Euclidean MMD for the saddle point problem $\min_{x \in \mathbb{R}} \max_{y \in \mathbb{R}} \frac{\alpha}{2} x^2 + (x - 1)(y - 1) - \frac{\alpha}{2} y^2$.

### D.6 BOUNDING THE GAP

Below we show how Theorem 3.4 can be used to guarantee linear convergence of the gap.

**Proposition D.8.** *Suppose the assumptions of Theorem 3.4 hold. Moreover, assume that $g$ is twice continuously differentiable over $\operatorname{int} \operatorname{dom} \psi$ and $\mathcal{Z}$ is bounded. In that case, there exists a constant $C$ and a time step $t'$ such that, for any $t \geqslant t'$,*

$$\theta_{gap}(z_t) = \sup_{z \in \mathcal{Z}} \langle F(z_t) + \alpha \nabla g(z_t), z_t - z \rangle \leqslant C \left( \sqrt{\frac{1}{1 + \eta \alpha}} \right)^{t - t'} \sqrt{B_\psi(z_*; z_{t'})}.$$

*Proof.* By Theorem 3.4 we know $z_t \to z_*$ where $\{z_t\}_{t \geqslant 1} \cup \{z_*\} \subseteq \operatorname{int} \operatorname{dom} \psi$. Therefore, we have that $\{z_t\}_{t \geqslant 1} \cup \{z_*\}$ is eventually within a closed ball centered at $z_*$. That is, there exists $t'$ and a closed ball $B$ such that $\{z_t\}_{t \geqslant t'} \cup \{z_*\} \subseteq B \subseteq \operatorname{int} \operatorname{dom} \psi$. Since $B$ is compact and $\nabla^2 g$ is continuous over $B$, we have that $\nabla^2 g(z)$ is bounded on $B$. Therefore, there exists $L_B$ such that $\|\nabla g(z') - \nabla g(z)\|_* \leqslant L_B \|z - z'\|$ for any $z, z' \in B$. Setting $G = F + \nabla g$, we have that, for any $z, z' \in B$, $\|G(z) - G(z')\|_* \leqslant \tilde{L} \|z - z'\|$ for $\tilde{L} = L + L_B$.

Then for any $z \in \mathcal{Z}, t \geqslant t'$, denoting $x_*$ as the solution to $\operatorname{VI}(\mathcal{Z}, G)$, we have

$$
\begin{aligned}
\langle G(z_t), z_t - z \rangle &= \langle G(z_*), z_t - z \rangle + \langle G(z_t) - G(z_*), z_t - z \rangle \\
&= \underbrace{\langle G(z_*), z_* - z \rangle}_{\leqslant 0} + \langle G(z_*), z_t - z_* \rangle + \langle G(z_t) - G(z_*), z_t - z \rangle \\
&\leqslant \|G(z_*)\|_* \|z_t - z_*\| + \tilde{L} \|z_t - z_*\| \|z_t - z\| \\
&\leqslant \left( \|G(z_*)\|_* + \tilde{L} D \right) \|z_t - z_*\| \\
&\leqslant C \sqrt{B_\psi(z_*; z_t)} \\
&\leqslant C \left( \sqrt{\frac{1}{1 + \eta \alpha}} \right)^{t - t'} \sqrt{B_\psi(z_*; z_{t'})}
\end{aligned}
$$

where $D$ is such that $\max_{z, z' \in \mathcal{Z}} \|z - z'\| \leqslant D$ and $C = \|G(z_*)\|_* + \tilde{L} D$. The first inequality is by the generalized Cauchy-Schwarz inequality and the Lipschitz property of $G$. The second inequality is by boundedness of $\mathcal{Z}$. The third inequality is by the fact that $B_\psi(z_*; z_t) \geqslant \frac{1}{2} \|z_* - z_t\|^2$. The fourth inequality is by applying Theorem 3.4 inductively. $\qquad \square$

Note that we have the following well-known inequality between the saddle-point gap

$$\xi(x, y) = \max_{\bar{y} \in \mathcal{Y}} \alpha g_1(x) + f(x, \bar{y}) - \alpha g_2(\bar{y}) - \min_{\bar{x} \in \mathcal{X}} \alpha g_1(\bar{x}) + f(\bar{x}, y) - \alpha g_2(y)$$

and

$$\theta_{gap}(z) = \sup_{\bar{z} \in \mathcal{Z}} \langle F(z) + \alpha \nabla g(z), z - \bar{z} \rangle,$$

as shown below:

$$\xi(x,y) = \max_{\bar{y} \in \mathcal{Y}} \alpha g_1(x) + f(x,\bar{y}) - \alpha g_2(\bar{y}) - \min_{\bar{x} \in \mathcal{X}} \alpha g_1(\bar{x}) + f(\bar{x},y) - \alpha g_2(y)$$

$$= \alpha g_1(x) + f(x,y') - \alpha g_2(y') - (\alpha g_1(x) + f(x,y) - \alpha g_2(y))$$

$$+ (\alpha g_1(x) + f(x,y) - \alpha g_2(y)) - (\alpha g_1(x') + f(x',y) - \alpha g_2(y)) \quad \text{for some pair } (x',y') \in \mathcal{X} \times \mathcal{Y}$$

$$\leqslant \langle -\nabla f_y(x,y) + \alpha \nabla g_2(y), y - y' \rangle + \langle \nabla f_x(x,y) + \alpha \nabla g_1(x), x - x' \rangle$$

$$= \langle F(z) + \alpha \nabla g, z - z' \rangle \text{ for } z = (x,y) \text{ and } z' = (x',y')$$

$$\leqslant \theta_{gap}(z).$$

Therefore Proposition D.6 gives a guarantee on the saddle-point gap $\xi(x,y)$.

## E  MMD FOR LOGIT-AQRES AND MINIMAXENT EQUILIBRIA

By Proposition D.2, the MMD update equation 6, restated below, has fixed points corresponding to the solutions of $\text{VI}(\mathcal{Z}, F + \nabla \psi)$:

$$z_{t+1} = \arg \min_{z \in \mathcal{Z}} \eta \left( \langle F(z_t), z \rangle + \alpha \psi(z) \right) + B_\psi(z; z_t).$$

If $\mathcal{Z}$ is the cross-product of policy spaces for both players (cross product of sets of behavioral-form policies) and $\psi$ is the sum of negative entropy over all decision points (information states), and $F$ includes the the negative q-values for both players, then the iteration above reduces to

$$\pi_{t+1}(h_i) \propto [\pi_t(h_i) e^{\eta q_{\pi_t}(h_i)}]^{1/(1+\eta\alpha)}$$

with fixed points corresponding to

$$\forall i, \forall h_i, \pi_i(h_i) \in \arg \max_{\pi_i'(h_i)} \mathbb{E}_{A \sim \pi_i'(h_i)} \left[ q_\pi(h_i, A) + \alpha \mathcal{H}(\pi_i'(h_i)) \right]$$

or, equivalently, the solution to $\text{VI}(\mathcal{Z}, F + \nabla \psi)$, which corresponds to a logit-AQRE. If $F$ includes the negative MiniMaxEnt $Q$-values for both players, the fixed point instead corresponds to a MiniMaxEnt equilibrium.

## F  EXPERIMENTAL DOMAINS

For our experiments with normal-form games, we used No-Press Diplomacy stage games. No-Press Diplomacy is a seven-player Markov game in which players compete to conquer Europe. Because the game is a Markov game (which means that the game is fully observable but that the players move simultaneously), each turn of the game resembles a normal-form game. We constructed the normal-form games that we used for our experiments by querying an open source value function (Bakhtin et al., 2021) in different circumstances for a two-player variant of the game, similarly to Zhang et al. (2022). These games have payoff matrices of shape $(50, 50)$ (game A), $(35, 43)$ (game B), $(50, 50)$ (game C), and $(4, 4)$ (game D). We normalized the payoffs of each game to $[0, 1]$.

For our extensive-form games, we used the implementations of Kuhn Poker, 2x2 (and also 3x3) Abrupt Dark Hex, 4-Sided Liar's Dice, and Leduc Poker provided by OpenSpiel (Lanctot et al., 2019). Kuhn poker (Kuhn, 1951) is a simplified poker game with three cards (J, Q, K). It has 54 non-terminal histories (not counting chance nodes). Abrupt Dark Hex is a variant of the classical board game Hex (Bakst & Gardner, 1962). In Hex, two players take turns placing stones onto a board. One player's goal is to create a path of its stones connecting the east end of the board with the west end, while the other player's goal is to do the same with the north end and south end. Dark Hex is a variant in which players cannot see where their opponents are placing stones. Abrupt Dark Hex is a variant of Dark Hex in which placing a stone in an occupied position results in a loss of turn. The prefix $n$x$n$ describes the size of the board. 2x2 Abrupt Dark Hex has 471 non-terminal histories. 3x3 Abrupt Dark Hex has too many non-terminal histories to enumerate on our hardware. Liar's Dice (Ferguson & Ferguson, 1991) is a dice game in which players privately roll dice and place

bids based on the observed outcomes, similarly to poker games. The prefix $n$-sided means that the players play with 4-sided dice. 4-Sided Liar's Dice has 8176 non-terminal histories (not counting chance nodes). Leduc Poker (Southey et al., 2005) is a small poker game with three card values (J, Q, K), each of which have two instances in the deck. It has 9300 non-terminal histories non-terminal histories (not counting chance nodes).

For our single-agent deep RL experiments, we use three Atari games (Bellemare et al., 2013) and three Mujoco games (Todorov et al., 2012). We selected these games because Huang et al. (2022) used them to benchmark an open source implementation of PPO.

## G   QRE EXPERIMENTS

### G.1   FULL FEEDBACK QRE CONVERGENCE DIPLOMACY

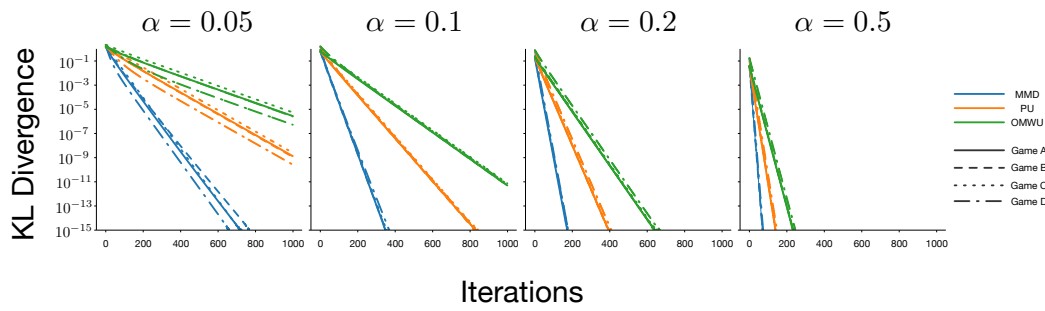

Figure 6: Solving for normal-form QREs in Diplomacy stage games with full feedback.

We perform various QRE experiments under full feedback for Diplomacy stage games. Full feedback means that each player outputs a fully specified policy and receives its exact Q-values (given both players' policies) as feedback. Both players then perform the update

$$\pi_{t+1}(h_i) \propto [\pi_t(h_i) e^{\eta q_{\pi_t}(h_i)}]^{1/(1+\eta\alpha)}.$$

For our experiments, we set $\eta = \alpha$ (for each $\alpha$) for MMD, which is the maximal value that retains a linear convergence guarantee for normal-form games with a max payoff magnitude of one. For PU and OMWU (Cen et al., 2021), we also used the maximal values that guarantee linear convergence. We solved for the QRE for each game using Ling et al.'s Newton's method approach. We show iterations on the $x$-axis and KL(solution, iterate) on the $y$-axis. We count each query to the oracle as an iterate, meaning that OMWU uses two iterates for every update (contrasting MMD and PU, which only use one).

The results of the experiment, found in Figure 6, show that all three algorithms converge linearly with faster rates for larger values of alpha, as is guaranteed by theory. We find that, for our Diplomacy games, MMD converges faster than PU and OMWU. However, we found that all three algorithms also exhibited faster convergence with larger than theoretically allowed stepsizes.

### G.2   BLACK BOX QRE CONVERGENCE DIPLOMACY

Our second set of experiments examine convergence to QREs for our Diplomacy stage games with black box feedback. In this context, black box feedback means that each player $i$ outputs an action $A_i$ sampled from its current policy and that player $i$ receives $\mathcal{R}(\cdot, A_i, A_{-i})$ (but not $A_{-i}$) as feedback. One way to approach such a setting is to construct an unbiased estimate of the exact Q-values. Letting $r$ be the observed reward

$$\hat{q}_t(a_i) = \begin{cases} r/\pi_t(a_i) & \text{if } A_i = a_i \\ 0 & \text{otherwise} \end{cases}$$

is such an estimate. To see that this is true, observe

$$\mathbb{E}[\hat{q}_t(a_i) \mid \pi_t] = \mathbb{E}_{A_{-i} \sim \pi_t} \left[ \pi_t(a_i) \cdot \frac{\mathcal{R}(\cdot, a_i, A_{-i})}{\pi_t(a_i)} + \sum_{a_i' \neq a_i} \pi_t(a_i') \cdot 0 \right]$$

$$= \mathbb{E}_{A_{-i} \sim \pi_t} \left[ \pi_t(a_i) \cdot \frac{\mathcal{R}(\cdot, a_i, A_{-i})}{\pi_t(a_i)} \right]$$

$$= \mathbb{E}_{A_{-i} \sim \pi_t} \mathcal{R}(\cdot, a_i, A_{-i})$$

$$= q_t(a_i).$$

In Figure 7, we show results for each of MMD, PU and OMWU, with the exact Q-values $q_t$ replaced by the unbiased estimates $\hat{q}_t$. For each algorithm, the stepsize at iteration $t$ was set to be equal to the maximal step size for which there exists an exponential convergence guarantee divided by $10\sqrt{t}$. In other words,

$$\eta_t = \frac{\eta}{10\sqrt{t}}.$$

Each line is an average over 30 runs. The bands depict estimates of 95% confidence intervals computed using bootstrapping. Although none of the algorithms possess existing black box convergence guarantees, we observe that they all exhibit convergent behavior empirically. In terms of convergence speed, we observe that MMD compares favorably to PU and OMWU for $\alpha \in \{0.05, 0.1, 0.2\}$; however, for $\alpha = 0.5$, OMWU performed the best, with the exception of game D. It is likely that all algorithms could achieve better performance, as we did not perform much hyperparameter tuning.

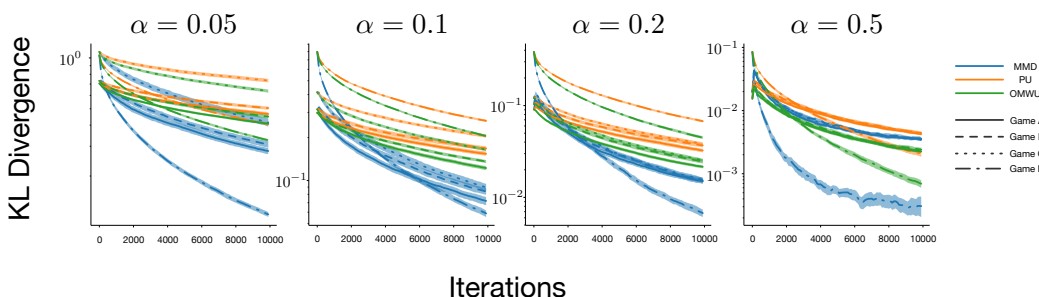

Figure 7: MMD, PU, and OMWU applied to Diplomacy stage games for QRE finding with black box sampling.

We also investigate the performance of other methods for estimating Q-values for the black box setting. One such method uses an unbiased baseline to reduce variance (Schmid et al., 2019; Davis et al., 2020). The premise of this approach is the idea that any quantity that is zero in expectation can be subtracted from an unbiased Q-value estimate without introducing bias. As a result, if the quantity is correlated with the estimator, subtracting it from the estimate can reduce variance "for free". We call this quantity a baseline. For our baseline, we used

$$b_t(a_i) = \begin{cases} \tilde{q}_t(a_i)/\pi_t(a_i) - \tilde{q}_t(a_i) & \text{if } a_i = A_i \\ -\tilde{q}_t(a_i) & \text{otherwise.} \end{cases}$$

By a similar argument as above, this quantity is zero in expectation

$$\mathbb{E}[b_t(a_i) \mid \pi_t] = \pi_t(a_i) \cdot (\tilde{q}_t(a_i)/\pi_t(a_i) - \tilde{q}_t(a_i)) - \sum_{a_i' \neq a_i} \pi_t(a_i') \cdot \tilde{q}_t(a_i)$$

$$= \tilde{q}_t(a_i) - \pi_t(a_i)\tilde{q}_t(a_i) - (1 - \pi_t(a_i)) \cdot \tilde{q}_t(a_i)$$

$$= \tilde{q}_t(a_i) - \tilde{q}_t(a_i)$$

$$= 0.$$

Also, if $\tilde{q}$ is close to $q$, our baseline will be correlated with $\hat{q}$. Thus, it satisfies our desired criteria. For $\tilde{q}$, we used a running estimate of the reward observed after selecting action $a_i$. Specifically, every time action $a_i$ was selected, we updated

$$\tilde{q}_t(a_i) = (1 - \tilde{\eta})\tilde{q}_t(a_i) + \tilde{\eta}r.$$

We used $\tilde{\eta} = 1/2$, inspired by Schmid et al. (2019).

We also investigated the use of *biased* Q-value estimates, as this is the setting that corresponds with function approximation. For this approach, we plugged in $\tilde{q}$, as computed above, instead of the exact Q-values $q$.

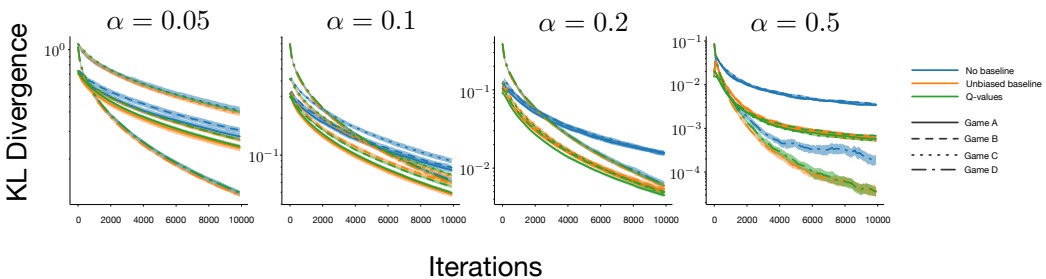

Figure 8: MMD with no baseline, an unbiased baseline, and (biased) Q-values applied to diplomacy stage games for QRE finding with black box sampling.

We show the results of the experiment if Figure 8. The column shows the temperature for the QRE. The y-axis shows the KL divergence to the corresponding logit-QRE. The x-axis shows the number of iterations. For each algorithm, the step size at iteration $t$ was set to be equal to the maximal step size for which there exists an exponential convergence guarantee divided by $10\sqrt{t}$. Each line is an average over 30 runs. The bands depict estimates of 95% confidence intervals computed using bootstrapping. Overall, we find that both using unbiased baselines and biased Q-value estimates appears to improve convergence speed.

### G.3 FULL FEEDBACK QRE CONVERGENCE EFGS

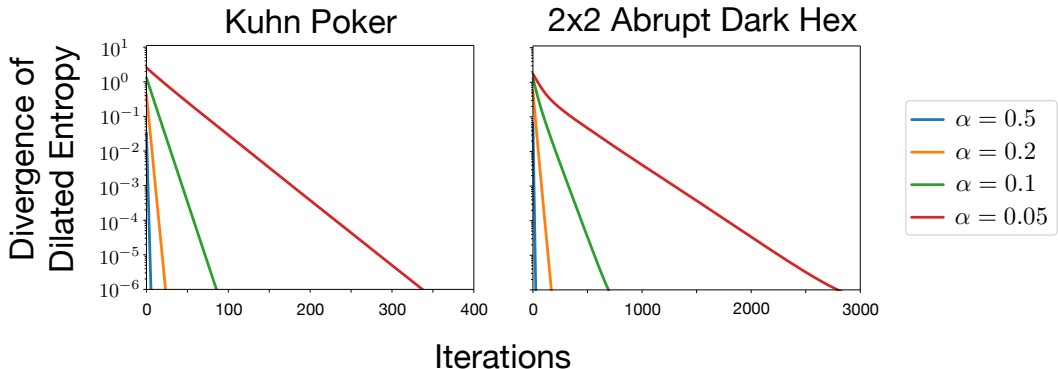

Figure 9: Convergence for normal-form QREs in EFGs, measured by divergence.

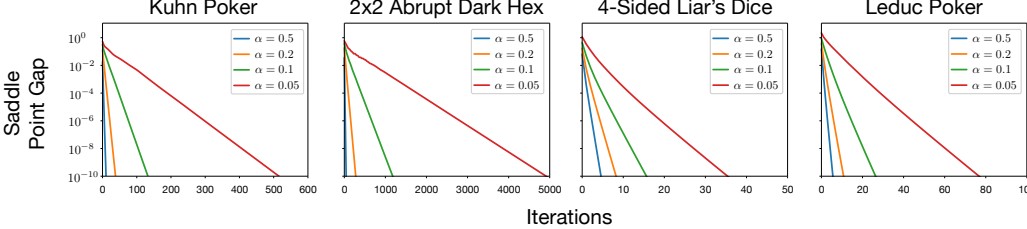

Figure 10: Convergence for normal-form QREs in EFGs, measured by saddle point gap.

We perform several experiments for solving reduced normal-form logit QREs by using MMD over the sequence form with dilated entropy. We use the descent-ascent updates

$$x_{t+1} = \arg\min_{x \in \mathcal{X}} \eta\left(\langle \nabla_{x_t} f(x_t, y_t), x \rangle + \alpha\psi_1(x)\right) + B_{\psi_1}(x; x_t),$$
$$y_{t+1} = \arg\max_{y \in \mathcal{Y}} \eta\left(\langle \nabla_{y_t} f(x_t, y_t), y \rangle - \alpha\psi_2(y)\right) - B_{\psi_2}(y; y_t).$$

The method is full feedback since $\nabla_{x_t} f(x_t, y_t) = Ay_t$ and $\nabla_{y_t} f(x_t, y_t) = A^\top x_t$, where $A$ is the sequence form payoff matrix. Note in the normal form setting $-Ay_t$ and $A^\top x_t$ are the Q-values for both players and the algorithm is the same as described in Section G.1. We set the stepsize to be $\eta = \alpha/(\max_{ij} |A_{ij}|)^2$, the largest possible allowed from Theorem 3.4. For more details on the sequence form algorithm, see Section C.3.

For Kuhn Poker and 2x2 Abrubt Dark Hex, we used Gambit (McKelvey, Richard D., McLennan, Andrew M., and Turocy, Theodore L., 2016; Turocy, 2005) to compute the reduced normal-form QRE. We check the convergence of MMD by plotting the sum of Bregman divergences with respect to dilated entropy $B_\psi(z_*; z_t) = B_{\psi_1}(x_*; x_t) + B_{\psi_1}(y_*; y_t)$, with respect to the solution $z_* = (x_*, y_*)$. As predicted by Theorem 3.4 we observer linear convergence with faster convergence for larger values of $\alpha$.

For 4-Sided Liar's Dice and Leduc Poker, the games were too large for Gambit (McKelvey, Richard D., McLennan, Andrew M., and Turocy, Theodore L., 2016; Turocy, 2005) to compute the reduced normal-form QRE on our hardware. Therefore, we check the convergence of MMD by plotting the saddle point gap $\xi(x_t, y_t)$ of the min max problem given by Ling et al. (2018),

$$\xi(x_t, y_t) = \max_{\bar{y} \in \mathcal{Y}} \alpha\psi_1(x_t) + x_t^\top A\bar{y} - \alpha\psi_2(\bar{y}) - \min_{\bar{x} \in \mathcal{X}} \alpha\psi_1(\bar{x}) + \bar{x}^\top Ay_t - \alpha\psi_2(y_t).$$

Theorem 3.4 guarantees that the gap will converge to zero. Note the gap is zero if and only if at the solution and, by Proposition D.6, the gap is also guaranteed to converge linearly. In both 4-Sided Liar's Dice and Leduc Poker we observe linear convergence of the gap, with faster convergence for larger values of $\alpha$. Additionally, due to the $O(\log(t))$ regret bound from Duchi et al. (2010), we have that the gap is guaranteed to converge at a rate of $O\left(\frac{\log(t)}{t}\right)$ for the average iterates of both players.

### G.4 FULL FEEDBACK AQRE CONVERGENCE EFGS

Next, we investigate whether MMD can be made to converge to AQREs in extensive-form games. For these experiments we applied MMD in behavioral form, as described in Section E. Specifically, we computed $q_t(h_i)$ for each player $i$ and each information state $h_i$. Then, we applied the update rule

$$\pi_{t+1}(h_i) \propto [\pi_t(h_i) e^{\eta q_{\pi_t}(h_i)}]^{1/(1+\eta\alpha)}.$$

for each player $i$ and information state $h_i$. For each setting, we used

$$\eta = \frac{\alpha}{10}.$$

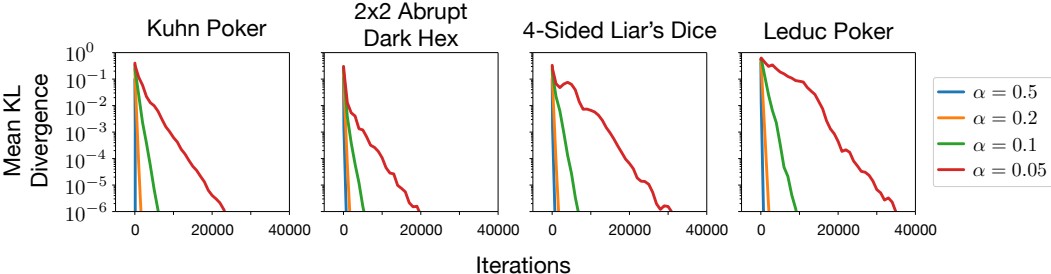

Figure 11: Solving for AQREs in EFGs.

We show the results in Figure 11. We measure convergence against solutions computed using Gambit (McKelvey, Richard D., McLennan, Andrew M., and Turocy, Theodore L., 2016; Turocy, 2010).

Despite a lack of proven convergence guarantees, we observe that MMD converges to the AQRE in each game, for each temperature. While the convergence is not monotonic, it is roughly linear over large time scales.

# H    EXPLOITABILITY EXPERIMENTS

Next, we investigate the convergence of MMD as a Nash equilibrium solver. To induce convergence, in most of our experiments, we anneal the temperature of the regularization over time.

## H.1    FULL FEEDBACK NASH CONVERGENCE DIPLOMACY

In our full feedback Nash convergence Diplomacy experiments, we used

$$\eta = \frac{1}{10}, \alpha_t = \frac{1}{5\sqrt{t}}.$$

We show the results of the experiment in Figure 12. Over short iteration horizons, we observe that CFR tends to outperform MMD. However, for longer horizons, we find that MMD tends to catch up with CFR. In game D, the qualitatively different behavior is likely to due the fact that the Nash equilibrium is a pure strategy, unlike the Nash equilibria of the first three games, which are mixed.

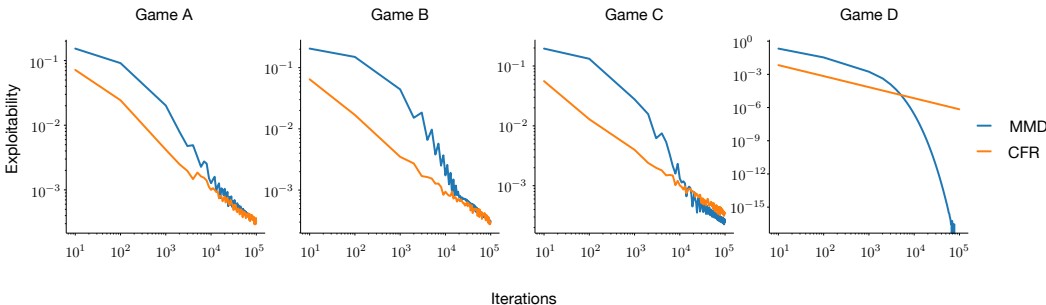

Figure 12: MMD and CFR applied to diplomacy stage games for computing Nash equilibria.

## H.2    BLACK BOX NASH CONVERGENCE DIPLOMACY

For our black box Nash convergence experiments, we compare against the "opponent on-policy" variant of Monte Carlo CFR (Lanctot et al., 2009). In this variant, the two players alternate between an updating player and an on-policy player. The updating player plays off-policy according to a policy that provides sufficiently large support to each action (in our Diplomacy experiments we used a uniform policy). The advantage to this setup is that it guarantees that the updating player will receive bounded gradients, which is necessary for Monte Carlo CFR's convergence proof. In contrast, we show results for an on-policy Monte Carlo variant of MMD, despite the fact that this causes unbounded gradients. This is not a fair comparison in the sense that the same "opponent on-policy" setup is equally applicable to MMD and would keep the gradients bounded, whereas the "on-policy" version of Monte Carlo CFR does not converge. We made this decision because the on-policy Monte Carlo variant of MMD is simpler and more elegant. Nevertheless, we believe that the "opponent on-policy" version of MMD remains an interesting direction for future, and would very possibly yield faster convergence.

We again investigated three ways of estimating Q-values. For our unbiased estimator with no baseline we used

$$\eta_t = \frac{1}{5\sqrt{t}}, \alpha_t = \frac{20}{\sqrt{t}}$$

for games A, B, and C, and

$$\eta_t = \frac{1}{\sqrt{t}}, \alpha_t = \frac{1}{\sqrt{t}}$$

for game D. For our unbiased estimator with baseline, we used

$$\eta_t = \frac{1}{10\sqrt{t}}, \alpha = \frac{10}{\sqrt{t}}$$

for games A, B, and C, and

$$\eta_t = \frac{1}{\sqrt{t}}, \alpha_t = \frac{1}{\sqrt{t}}$$

for game D. For our biased estimator, we used

$$\eta_t = \frac{1}{5\sqrt{t}}, \alpha_t = \frac{2}{\sqrt{t}}$$

for games A, B, and C, and

$$\eta_t = \frac{1}{\sqrt{t}}, \alpha_t = \frac{1}{\sqrt{t}}$$

for game D. We show results in Figure 15, with averages across 30 runs and estimates of 95% confidence intervals computed from bootstrapping.

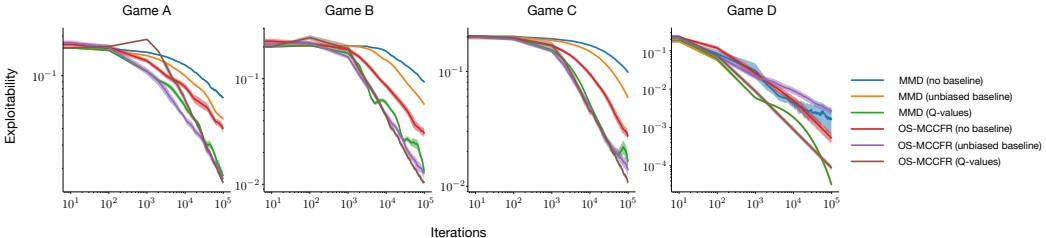

Figure 13: MMD and OS-MCCFR applied to diplomacy stage games for computing Nash equilibria with black box sampling.

For MMD, we observe that biased Q-value estimates generally perform best, followed by an unbiased estimate with baseline, followed by an unbiased estimate without baseline, except on game D, where the unbiased baseline performs similarly to biased Q-value estimates. We also find that CFR tends to follow this trend, though the difference between biased Q-value estimates and an unbiased baseline is less pronounced, except on game D, where the unbiased baseline performs poorly. Between MMD and CFR, CFR tends to perform better on an estimator-to-estimator basis in games A, B and C, though MMD is relatively competitive with CFR under biased Q-value estimates. For game D, we observe that this comparison is more favorable for MMD than the other games.

### H.3 FULL FEEDBACK NASH CONVERGENCE EFGS

For our full feedback Nash convergence EFG experiments, we examined two variants of MMD. The first, which we call unweighted MMD, corresponds with the version tested in the AQRE experiments

$$\pi_{t+1}(h_i) \propto (\pi_t(h_i) e^{\eta_t q_{\pi_t}(h_i)})^{1/(1+\eta_t \alpha_t)}.$$

The second, which we call weighted MMD, uses

$$\pi_{t+1}(h_i) \propto (\pi_t(h_i) e^{\mathcal{P}^{\pi_t}(h_i)\eta_t q_{\pi_t}(h_i)})^{1/(1+\mathcal{P}^{\pi_t}(h_i)\eta_t \alpha_t)}.$$

In other words, it weights the stepsize of the update by the probability of reaching that information state under the current policy. We test this variant because it corresponds with a "determinized" version of black box sampling for temporally extended settings.

For unweighted MMD, we used

$$\eta_t = \frac{1}{\sqrt{t}}, \alpha_t = \frac{1}{\sqrt{t}}$$

for Kuhn Poker,

$$\eta_t = \frac{1}{\sqrt{t}}, \alpha_t = \frac{1}{\sqrt{t}},$$

for 2x2 Dark Hex,

$$\eta_t = \frac{2}{\sqrt{t}}, \alpha_t = \frac{1}{\sqrt{t}},$$

for 4-Sided Liar's dice, and

$$\eta_t = \frac{1}{\sqrt{t}}, \alpha_t = \frac{5}{\sqrt{t}}$$

for Leduc Poker.

For weighted MMD, we used

$$\eta_t = \frac{2}{\sqrt{t}}, \alpha_t = \frac{2}{\sqrt{t}}$$

for Kuhn Poker,

$$\eta_t = \frac{1}{\sqrt{t}}, \alpha_t = \frac{1}{\sqrt{t}}$$

for 2x2 Abrupt Dark Hex,

$$\eta_t = \frac{100}{\sqrt{t}}, \alpha_t = \frac{2}{\sqrt{t}}$$

for 4-Sided Liar's Dice, and

$$\eta_t = \frac{500}{\sqrt{t}}, \alpha_t = \frac{10}{\sqrt{t}}$$

for Leduc Poker. Note that larger stepsize values are required for weighted MMD to achieve competitive performance because, otherwise, the reach probability weighting would make updates at the bottom of the tree very small.

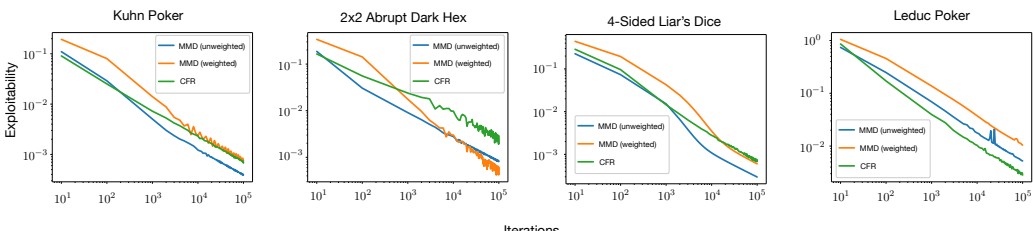

Figure 14: MMD (unweighted) and MMD (weighted by reach probability) compared against CFR across standard OpenSpiel games.

We show the results of our experiments in Figure 14. We find that both weighted MMD and unweighted MMD exhibit convergent behavior. Furthermore, they converge at rates comparable with CFR on average across the games.

### H.4 BLACK BOX NASH CONVERGENCE EFGS

For our black box Nash convergence EFG experiments, we used the Monte Carlo CFR implementation in OpenSpiel (Lanctot et al., 2019), which uses an update policy with a $0.4$ weight on the current policy and a $0.6$ weight on the uniform policy. For MMD, we used the sampling version of weighted MMD, meaning that the information states touched during the trajectory are updated with the full stepsize, while information not touched during the trajectory are not updated. For Kuhn Poker, we used

$$\eta_t = \frac{1}{10\sqrt{t}}, \alpha_t = \frac{50}{\sqrt{t}}.$$

For 2x2 Abrupt Dark Hex, we used

$$\eta_t = \frac{7}{20\sqrt{t}}, \alpha = \frac{50}{\sqrt{t}}.$$

For 4-Sided Liar's Dice, we used

$$\eta_t = \frac{2}{\sqrt{t}}, \alpha_t = \frac{200}{\sqrt{t}}.$$

For Leduc Poker, we used

$$\eta_t = \frac{1}{\sqrt{t}}, \alpha_t = \frac{300}{\sqrt{t}}.$$

Noting again that the caveats about comparing on-policy MMD to opponent on-policy Monte Carlo CFR also apply here, we present the results in Figure 14. Results are averaged across 30 runs and shown with 95% confidence intervals estimated from bootstrapping. As in the normal-form experiments, we find that Monte Carlo CFR generally outperforms MMD for unbiased gradient estimates with no baseline.

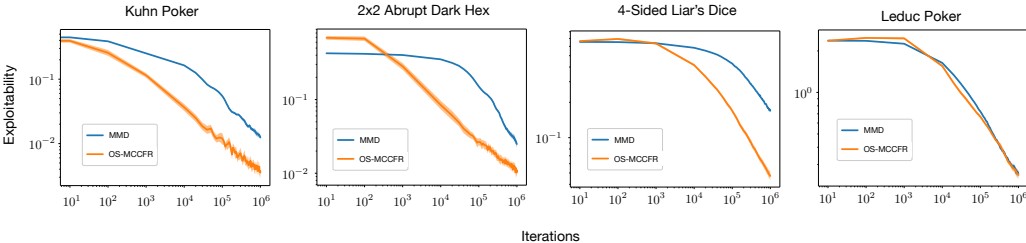

Figure 15: MMD compared against OS-MCCFR across standard OpenSpiel games with black box sampling.

## H.5 MOVING MAGNET

Next, we investigate using a moving magnet, rather than an annealing temperature, to induce convergence to a Nash equilibrium. In the moving magnet setup, updates take the form

$$\pi_{t+1}(h_i) \propto [\pi_t(h_i)\rho_t(h_i)^{\eta\alpha}e^{\eta q_{\pi_t}(h_i)}]^{1/(1+\eta\alpha)},$$

where $\rho_t$ slowly trails behind $\pi_t$. In our experiment, we used

$$\rho_{t+1}(h_i) \propto \rho_t(h_i)^{1-\tilde{\eta}}\pi_{t+1}(h_i)^{\tilde{\eta}}.$$

For each game, we used $\alpha = 1$, $\eta = 0.1$, $\tilde{\eta} = 0.05$.

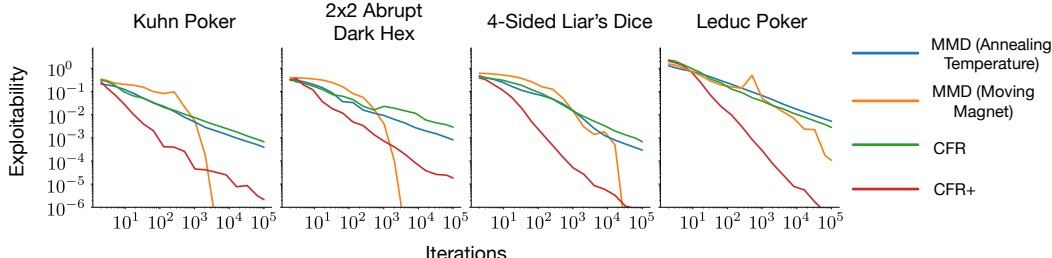

Figure 16: Comparing a moving magnet to an annealing temperature.

We show the results in Figure 16, compared against CFR and MMD with an annealing temperature (with the same hyperparameters as before). Encouragingly, we find that that moving the magnet behind the current iterate also appears to induce convergence. Furthermore, convergence may occur at a much faster rate than that which is induced by annealing the temperature.

## H.6 MAXENT AND MINIMAXENT OBJECTIVES

Next, we examine the convergence properties of other related objectives. We consider two different types. One involves an information state entropy bonus, wherein $\alpha\mathcal{H}(\pi_t(h_i))$ is added to the reward for player $i$ for reaching information state $h_i$. This corresponds with a maximum entropy objective in reinforcement learning (Ziebart et al., 2008); we call this objective MaxEnt. The second involves simultaneously giving an information state entropy bonus (like MaxEnt), while also penalizing the

player with the opponent's information state entropy. This second approach can be viewed as a modification of the first approach that makes the game zero-sum. It is the objective that was examined in Pérolat et al. (2021). We call this objective MiniMaxEnt.

For each algorithm, we used

$$\eta_t = \frac{1}{\sqrt{t}}, \alpha_t = \frac{1}{\sqrt{t}}$$

for Kuhn Poker,

$$\eta_t = \frac{1}{\sqrt{t}}, \alpha_t = \frac{1}{\sqrt{t}},$$

for 2x2 Dark Hex,

$$\eta_t = \frac{2}{\sqrt{t}}, \alpha_t = \frac{1}{\sqrt{t}},$$

for 4-Sided Liar's Dice, and

$$\eta_t = \frac{1}{\sqrt{t}}, \alpha_t = \frac{5}{\sqrt{t}}$$

for Leduc Poker.

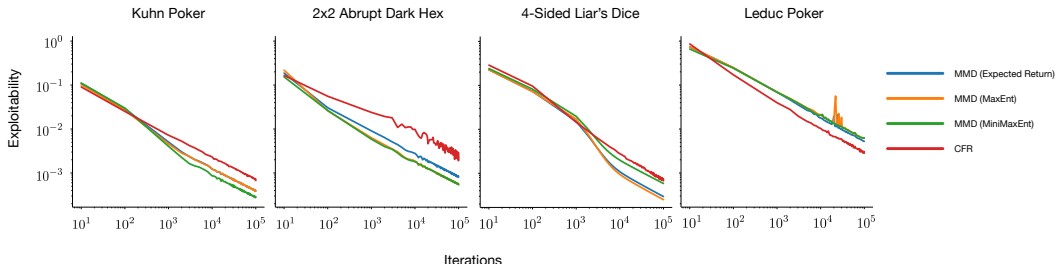

Figure 17: Comparing different objectives.

We show the results in Figure 17. We find that MMD exhibits convergent behavior with each of the objectives.

### H.7 EUCLIDEAN MIRROR MAP OVER LOGITS

Next, we examine an instantiation of MMD that optimizes the logits using a Euclidean mirror map ($\psi = \frac{1}{2}\|\cdot\|_2^2$), as discussed in Section D.5, rather than reverse KL regularization. The update rule for this approach is given by

$$z_{t+1}(h_i) = \arg\max_z \langle \nabla_w \mathbb{E}_{a\sim\text{softmax(w)}} q_{\pi_t}(h_i, a)|_{w=z_t(h_i)}, z\rangle - \frac{\alpha}{2}\|z - \zeta(h_i)\|^2 - \frac{1}{2\eta}\|z - z_t(h_i)\|^2$$

where $\pi_t(h_i) = \text{softmax}(z_t(h_i))$ and $\zeta$ is the magnet. The closed form is

$$z_{t+1}(h_i) = \frac{z_t(h_i) + \eta\nabla_w \mathbb{E}_{a\sim\text{softmax}(w)} q_{\pi_t}(h_i, a)|_{w=z_t(h_i)} + \alpha\eta\zeta(h_i)}{1 + \alpha\eta}.$$

We test the convergence of Euclidean MMD for Leduc poker, using

$$\eta_t = \frac{2}{\sqrt{t}}, \alpha_t = \frac{1}{\sqrt{t}}.$$

We show the results in Figure 18. We find that Euclidean MMD also exhibits convergence behavior in Leduc poker. However, convergence may be slower than the negative entropy variant.

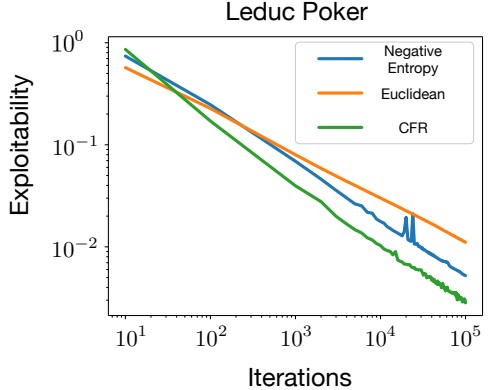

Figure 18: Comparing different mirror maps.

## H.8 MINIMAXENT EXPLOITABILITY WITH FIXED PARAMETERS

Next, we examine using MMD for the purposes of computing MiniMaxEnt equilibria. For each temperature $\alpha$, we used $\eta = \alpha/10$. We show the results in Figure 19. In the figure, convergence is measured in terms of exploitability in the entropy regularized game. Similarly to our AQRE results, we find that, although convergence is non-monotonic, the empirical rate appears roughly linear over long time scales. This is the first empirical demonstration of convergence in MiniMaxEnt exploitability in EFGs.

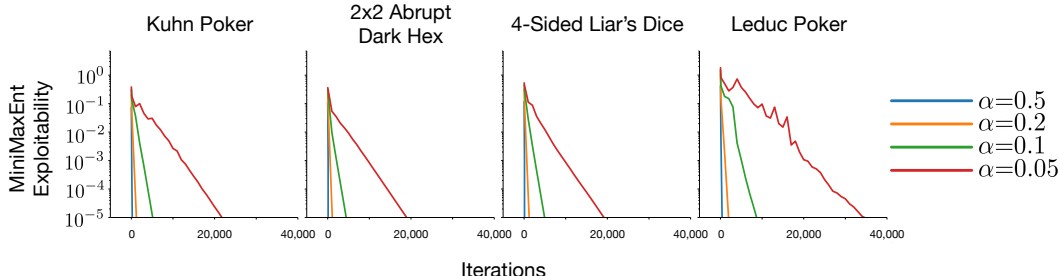

Figure 19: MiniMaxEnt exploitability with fixed parameter values.

## H.9 EXPLOITABILITY WITH FIXED PARAMETERS

In our AQRE and MiniMaxEnt experiments, we observed that MMD exhibits convergent behavior to AQREs and MiniMaxEnt equilibria with fixed parameter values. It follows that MMD can achieve relatively good exploitability values without using any scheduling. We show these results in Figure 20 and Figure 21.

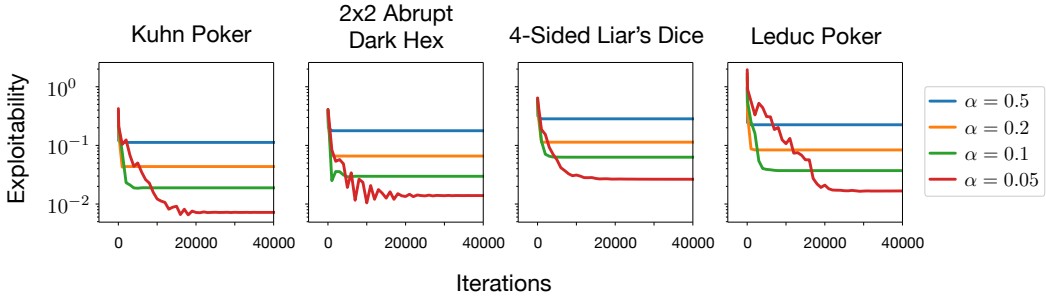

Figure 20: Convergence with fixed hyperparameter values under an expected return objective.

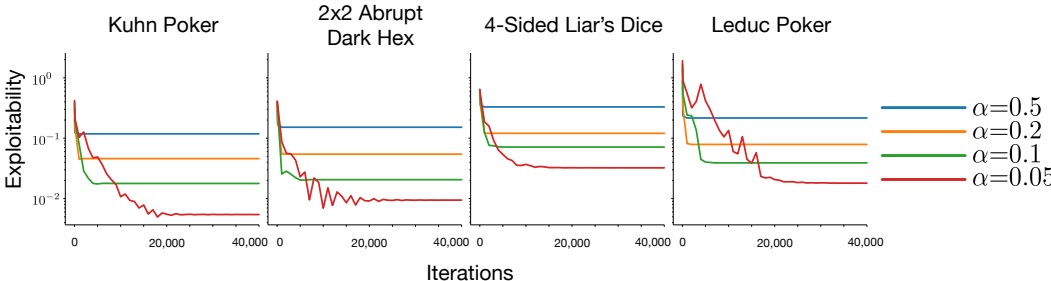

Figure 21: Convergence with fixed hyperparameter values under a MiniMaxEnt objective.

## I  ATARI AND MUJOCO EXPERIMENTS

For our single-agent deep RL experiments, we implemented MMD as a modification to Huang et al.'s implementation of PPO. For Atari, we added a reverse KL penalty with a coefficient $1/\eta = 0.001$; we kept the temperature as the default value set by Huang et al. (2022) ($\alpha = 0.01$). For Mujoco, we added a reverse KL penalty with a coefficient $1/\eta = 0.1$; we added an entropy bonus (Huang et al. (2022) do not use an entropy bonus) with a value of $\alpha = 0.0001$. Otherwise, for both Atari and Mujoco, the hyperparameters were set to those selected by Huang et al. (2022). We show the results again in Table 1 for convenience. The baseline results for PPO are copied directly from Huang et al. (2022). The exact numbers should be interpreted cautiously as they are averaged over only three runs, leaving high levels of uncertainty. That said, the results in Table 1 provide evidence that MMD can perform comparably to PPO. But, even without looking at empirical results, the idea that a deep form of MMD can perform comparably to PPO should not be surprising, as MMD can be implemented in a way that resembles PPO in many aspects.

Table 1: Atari and Mujoco results averaged over 3 runs, with standard errors.

|  | Breakout | Pong | BeamRider | Hopper-v2 | Walker2d-v2 | HalfCheetah-v2 |
|---|---|---|---|---|---|---|
| PPO | $409 \pm 31$ | $20.59 \pm 0.40$ | $2628 \pm 626$ | $2448 \pm 596$ | $3142 \pm 982$ | $2149 \pm 1166$ |
| MMD | $414 \pm 6$ | $21.0 \pm 0.00$ | $2549 \pm 524$ | $2898 \pm 544$ | $2215 \pm 840$ | $3638 \pm 782$ |

## J  DEEP MULTI-AGENT REINFORCEMENT LEARNING EXPERIMENTS

For our deep multi-agent reinforcement learning experiments, we implemented MMD as a modification to PPO, as implemented by RLlib (Liang et al., 2018). This involved changing the adaptive forward KL regularization to a constant reverse KL regularization and setting $\eta_t, \alpha_t$ according the following schedule

$$\eta_t = 0.05\sqrt{\frac{10 \text{ million}}{t}}, \alpha_t = 0.05\sqrt{\frac{10 \text{ million}}{t}},$$

where $t$ is the number of time steps—not the number of episodes. Otherwise, we used the default hyperparameters. We also show results for PPO, using RLlib's default hyperparameters. We ran these implementations in self-play using RLlib's OpenSpiel environment wrapper, modified to work with information states, rather than observations. For NFSP (Heinrich & Silver, 2016), we used the same hyperparameters as those found in the NFSP Leduc example in the OpenSpiel codebase. For the best response, we used the OpenSpiel's DQN best response code, without modifying any hyperparameters. We ran the best response for 10 million time steps and evaluated all match-ups over 2000 games (with each agent being the first-moving player in 1000).

There are two caveats to consider in interpreting these experiments. First, it is likely that RLlib's default PPO hyperparameters are generally stronger than the default hyperparameters for NFSP in the OpenSpiel. In this respect, the results we present may be unfair to NFSP. Second, RLlib's OpenSpiel wrapper does endow agents with knowledge about which actions are legal—instead, if an illegal action is selected, the agent is given a small penalty and a random legal action is executed. In contrast, OpenSpiel's implementation of NFSP uses information about the legal actions to perform masking.

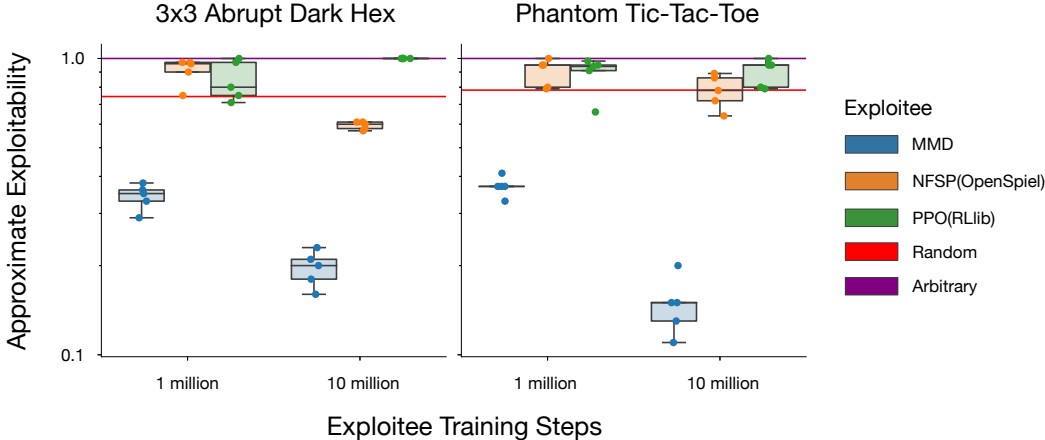

Figure 22: Approximate exploitability experiments.

In other words, MMD and PPO face a harder version of the game than NFSP faces. In this respect, the results we present are unfair to MMD and PPO.

We ran five seeds of each algorithm and checkpointed the parameters after both 1 million time steps and 10 million time steps. Because these games are too large to compute exact exploitability, we show results for DQN (Mnih et al., 2015) instances trained to best respond to each agent. For our DQN implementation, we used OpenSpiel's rl_response implementation. We did not modify any hyperparameters and ran DQN for 10 million time steps in all cases. We show results for the final DQN agent evaluated over 2000 games (1000 with DQN moving first and 1000 with DQN moving second) rounded to two decimal places. We also include results for a bot that selects actions uniformly at random (Random) and a bot that determinisically selects the first legal action (Arbitrary). These results of this experiment are presented in Figure 22.

In the games, the return for a win is 1 and the return for a loss is -1; in Phantom Tic-Tac-Toe, it is also possible to tie, in which case the return is 0. Thus, an approximate exploitability of 1 would mean that DQN defeats the agent 100% of the time, whereas an approximate exploitability of 0 would mean that DQN ties in expected value against the agent. As might be expected, we observe that playing an arbitrary deterministic policy (purple line) is perfectly exploitable by DQN in both games. In contrast, while playing uniformly at random (red line) is also highly exploitable, it is less so because of the randomization.

Among the three learning agents, one trend is that PPO with RLlib's default hyperparameters does not appear to decrease exploitability over time. This is not necessarily surprising, as RL algorithms do not generally converge in two-player zero-sum games. On the other hand, both NFSP with OpenSpiel hyperparameters and MMD exhibit clear downward trends over time. Again, this is not necessarily surprising, as both MMD and NFSP are designed with exploitability in mind.

Among the learning agents, in terms of raw value, MMD exhibits substantially stronger performance than the baselines. Indeed, even after 1 million time steps, every seed of MMD is less exploitable than any seed of the baselines after either 1 million or 10 million time steps. In contrast, the learning agent baselines do not substantially outperform uniform random play in Phantom Tic-Tac-Toe and only NFSP after 10 million time steps substantially outperforms uniform random play in 3x3 Abrupt Dark Hex.

We also show results for head-to-head matchups between the agents in Figure 23. For all learning agents, we use the 5 seeds that were trained for 10 million time steps. For matchups between learning agents, we ran each seed of each agent against each other (for a total of 25 matchups) for 2000 games (1000 with each agent moving first) and rounded to the nearest two decimal places. For matchups between learning agents and bots, we ran each seed of the learning agent against the bot (for a total of 5 matchups) for 2000 games (1000 with each agent moving first) and rounded to the nearest two decimal places. We show the results in Figure 23.

Each learning algorithm's results are denoted by an x-axis label; the hue denotes the opponent—not the agent being evaluated. Because the games are zero-sum matchup results are negations of each other. For example, in the 3x3 Abrupt Dark Hex column, the orange boxplot (i.e., opponent=NFSP) above the MMD label is the negation of the blue boxplot (i.e., opponent=MMD) above the NFSP label.

One observation from the results is that MMD outperforms the baselines and the bots uniformly across seeds. This is encouraging in the sense that having low approximate exploitability appears to lead to strong performance in head-to-head matchups. Like MMD, the NFSP seeds win head-to-head matchups against the bots. On the other hand, PPO exhibits much higher variance—it tends to lose against the bot selecting the first legal action in Phantom Tic-Tac-Toe, but tends to defeat it by the largest margin in 3x3 Abrupt Dark Hex.

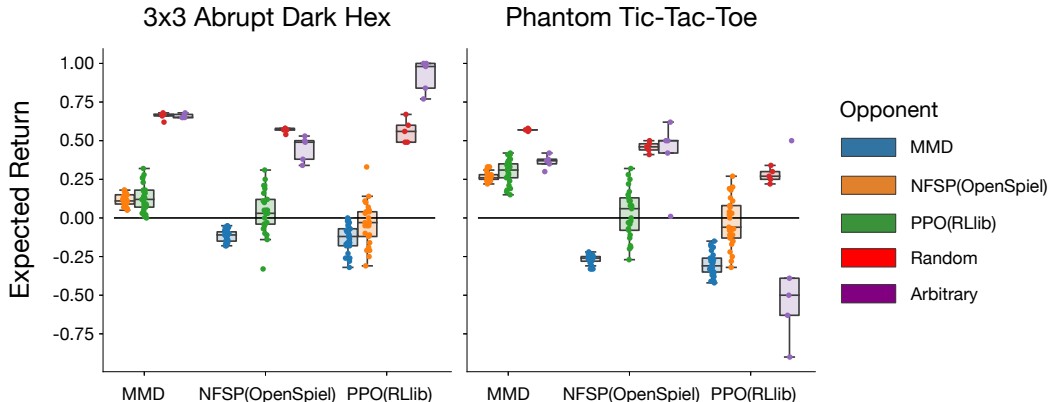

Figure 23: Head-to-head experiments.

## J.1 RESULTS AVERAGED ACROSS SEEDS

Table 2: Approximate exploitability and standard error in units of $10^{-2}$ over 5 seeds.

|                          | 3x3 Abrupt Dark Hex | Phantom Tic-Tac-Toe |
|--------------------------|:-------------------:|:-------------------:|
| First Legal Action Taker | $100 \pm 0$         | $100 \pm 0$         |
| Uniform Random           | $74 \pm 1$          | $78 \pm 0$          |
| PPO(1M steps)            | $85 \pm 6$          | $89 \pm 6$          |
| PPO(10M steps)           | $100 \pm 0$         | $90 \pm 4$          |
| NFSP(1M steps)           | $91 \pm 4$          | $95 \pm 1$          |
| NFSP(10M steps)          | $59 \pm 1$          | $78 \pm 5$          |
| MMD(1M steps)            | $34 \pm 2$          | $37 \pm 1$          |
| MMD(10M steps)           | $20 \pm 1$          | $15 \pm 1$          |

Table 3: Head-to-head expected return in 3x3 Abrupt Dark Hex for column player and standard error in units of $10^{-2}$.

|           | Arbitrary   | Random      | PPO         | NFSP        | MMD         |
|-----------|:-----------:|:-----------:|:-----------:|:-----------:|:-----------:|
| Arbitrary | 0           | $0 \pm 2$   | $92 \pm 5$  | $45 \pm 4$  | $66 \pm 1$  |
| Random    | $0 \pm 2$   | 0           | $56 \pm 3$  | $57 \pm 1$  | $66 \pm 1$  |
| PPO       | $-92 \pm 5$ | $-56 \pm 3$ | 0           | $4 \pm 3$   | $13 \pm 2$  |
| NFSP      | $-47 \pm 9$ | $-57 \pm 1$ | $-4 \pm 3$  | 0           | $12 \pm 1$  |
| MMD       | $-66 \pm 1$ | $-66 \pm 1$ | $-13 \pm 2$ | $-12 \pm 1$ | 0           |

Table 4: Head-to-head expected return in Phantom Tic-Tac-Toe for column player and standard error in units of $10^{-2}$.

|           | Arbitrary  | Random    | PPO       | NFSP      | MMD      |
|-----------|------------|-----------|-----------|-----------|----------|
| Arbitrary | 0          | $-22 \pm 2$ | $-28 \pm 24$ | $41 \pm 10$ | $36 \pm 2$ |
| Random    | $-22 \pm 2$ | 0         | $28 \pm 5$ | $46 \pm 2$ | $57 \pm 0$ |
| PPO       | $28 \pm 24$ | $-28 \pm 5$ | 0         | $3 \pm 3$ | $30 \pm 2$ |
| NFSP      | $-41 \pm 10$ | $-46 \pm 2$ | $-3 \pm 3$ | 0         | $27 \pm 1$ |
| MMD       | $-36 \pm 2$ | $-57 \pm 0$ | $-30 \pm 2$ | $-27 \pm 1$ | 0        |

## K    ADDITIONAL RELATED WORK

**Average Policy Deep Reinforcement Learning for Two-Player Zero-Sum Games** One class of deep reinforcement learning methods for two-player zero-sum games, which includes NFSP (Heinrich & Silver, 2016) and PSRO (Lanctot et al., 2017), scales oracle-based approaches (Brown, 1951; McMahan et al., 2003) by using single-agent reinforcement learning as a subroutine to compute approximate best responses. While this class of methods is very scalable, it can require computing many best responses (McAleer et al., 2021), making it very slow in some cases. MMD differs from this class in that it does not use a best response subroutine and in that it does not use averages over historical policies. Another class of methods, which includes deep CFR (Brown et al., 2019) and double neural CFR (Li et al., 2020), is motivated by scaling CFR (Zinkevich et al., 2007)—the dominant paradigm in tabular settings—to function approximation. Unfortunately, the sampling variant of CFR (Lanctot et al., 2009) requires importance sampling across trajectories, making straightforward extensions to stochasticity (Steinberger et al., 2020) difficult to apply to games with long trajectories, though more recent extensions may make progress toward resolving this issue (Gruslys et al., 2021). MMD differs from this class both in that it neither requires policy averaging nor importance sampling over trajectories.

## L    RELATIONSHIP TO KL-PPO AND MDPO

On the single-agent deep reinforcement learning side, MMD most closely resembles KL-PPO (Schulman et al., 2017) and MDPO (Tomar et al., 2020; Hsu et al., 2020).[8] KL-PPO uses the policy loss function

$$\mathbb{E}_t \left[ \frac{\pi(a_t \mid s_t)}{\pi_{\text{old}}(a_t \mid s_t)} \hat{A}_t + \alpha \mathcal{H}(\pi(s_t)) - \beta \text{KL}(\pi_{\text{old}}(s_t), \pi(s_t)) \right],$$

where $\hat{A}_t$ is an advantage function (a learned estimate of $q_{\pi_t}(s_t, a_t) - v_{\pi_t}(s_t)$). In expectation, the first term acts as $\langle \pi_t(s_t), q_{\pi_t}(s_t) \rangle$, which is the first term of MMD's loss function. The second term is the same entropy bonus as exists in MMD's loss function, using a uniform magnet with a negative entropy mirror map. However, unlike MMD, KL-PPO's KL regularization goes forward $\text{KL}(\pi_{\text{old}}(s_t), \pi(s_t))$. In contrast, MMD's KL regularization goes backward $\text{KL}(\pi(s_t), \pi_{\text{old}}(s_t))$. Hsu et al. (2020) investigated modifying KL-PPO to use reverse KL regularization instead of forward KL in Mujoco and found that the two yielded similar performance.

MDPO uses the policy loss function

$$\mathbb{E}_t \left[ \frac{\pi(a_t \mid s_t)}{\pi_{\text{old}}(a_t \mid s_t)} \hat{A}_t - \beta \text{KL}(\pi(s_t), \pi_{\text{old}}(s_t)) \right],$$

where $\hat{A}_t$ is the approximate advantage function (a learned estimate of $q_{\pi_t}(s_t, a_t) - v_{\pi_t}(s_t)$). In the context of a negative entropy mirror map and a uniform magnet, MDPO differs from MMD in that it does not necessarily include an entropy regularization term $\alpha \mathcal{H}(\pi(s_t))$; in the case that it does include such an entropy term, MDPO and MMD coincide.

MMD with a negative entropy mirror map and a uniform magnet takes the form

$$\mathbb{E}_t \left[ \frac{\pi(a_t \mid s_t)}{\pi_{\text{old}}(a_t \mid s_t)} \hat{A}_t + \alpha \mathcal{H}(\pi(s_t)) - \beta \text{KL}(\pi(s_t), \pi_{\text{old}}(s_t)) \right],$$

where $\beta$ acts as an inverse stepsize.

---

[8]Hsu et al. (2020) investigate MDPO under the name PPO reverse KL.

# M  FUTURE WORK

Our work opens up a multitude of important directions for future work. On the theoretical side, these directions include pursuing results for black box sampling, behavioral-form convergence, convergence with annealing regularization, convergence with moving magnets (Lin et al., 2017; Allen-Zhu, 2017), and relaxing the smoothness assumption to relative smoothness (Lu et al., 2018; Bauschke et al., 2017) while removing strong convexity assumptions. On the empirical side, these directions include constructing an adaptive mechanism for adapting the stepsize, temperature, and magnet (Badia et al., 2020; Fan & Xiao, 2022), pushing the limits of MMD as a deep RL algorithm for large scale 2p0s games, and investigating MMD as an optimizer for differentiable games (Goodfellow et al., 2014; Letcher et al., 2019).

