# OpenReview forum: "A Unified Approach to Reinforcement Learning, Quantal Response Equilibria, and Two-Player Zero-Sum Games"
_ICLR.cc/2023/Conference — ICLR 2023 poster_

### Official Review · Reviewer_8q9G · 2022-10-23

**Confidence:** 3
**Clarity, Quality, Novelty And Reproducibility:** The novelty of the algorithm and the …
**Correctness:** 4
**Technical Novelty And Significance:** 2
**Empirical Novelty And Significance:** 2
**Recommendation:** 5

**Strength And Weaknesses:**

Strength:
1.	This paper provides a unified perspective to view the RL problem and 2p0s games and proposes an algorithm to solve it.
2.	The theoretical results are strong and some results are attained for the first time as the authors claim.
Weaknesses:
1.	The algorithm seems not to be of novelty. Maybe the contributions mainly lie in the analysis.
2.	The paper is not well organized, and the problem is not highly motivated. The comparisons with existing and the idea behind the algorithm are not clearly presented.


**Summary Of The Paper:**

The paper proposes an algorithm, magnetic mirror descent (MMD), for RL problem and 2p0s games. Sound theoretical results are obtained for the algorithm. Empirically, MMD exhibits desirable properties as a tabular equilibrium solver, as a single-agent deep RL algorithm, and as a multi-agent deep RL algorithm.

**Summary Of The Review:**

The idea to deal with RL and games in a unified way is good, and the algorithm proposed enjoys linear convergence. However, it seems that the improved convergence rate compared with existing approach results from stronger assumption. The authors are expected to provide more insights of the algorithm and the novelty of the algorithm and its analysis scheme.

---

> ### Author Response · Authors · 2022-11-11
> **Author Comment**
>
>
> Thanks for your thoughtful review!
>
> > The algorithm seems not to be of novelty.
>
> We respectfully disagree that the algorithm is not novel. Algorithm 3.1 is a generalization of the non-Euclidean proximal gradient method from convex optimization to variational inequalities (similarly to how gradient descent ascent is a generalization of gradient descent). This generalization allows us to handle interesting problems, such as QREs in EFGs, that cannot be modeled as convex optimization problems.
>
> > Maybe the contributions mainly lie in the analysis.
>
> The main contributions are three fold:
> 1. We provide a novel convergence guarantee for the variational inequality literature with novel analysis that combines relative strong convexity (from convex optimization) and smooth monotone operators (from variational inequalities).
> 2. We attain the first linear convergence result for computing QREs in EFGs.
> 3. We show that an RL algorithm can perform competitively with CFR -- addressing a longstanding unresolved research direction in the imperfect information games community.
>
> > The paper is not well organized, and the problem is not highly motivated.
>
> If the reviewer has any suggestions on improving the organization, please let us know.
>
>
> > The novelty of the algorithm and the analysis method need to be highlighted.
>
> We agree that the novelty of the analysis can be better highlighted. We have modified the abstract, introduction, and Section 3, to better emphasize our algorithmic contribution to the variational inequality literature.   Given the limited space we chose to highlight the important contribution of our result within the QRE literature --- a new algorithm with linear convergence in EFGs, a feat not yet achieved by any other algorithm.
>
> > The comparisons with existing and the idea behind the algorithm are not clearly presented.
>
> We tried to extensively document the relationship between MMD and existing work in the related work section. If the reviewer has any suggestions on how would could make these comparisons clearer, please let us know.
>
> > The idea to deal with RL and games in a unified way is good, and the algorithm proposed enjoys linear convergence. However, it seems that the improved convergence rate compared with existing approach results from stronger assumption.
>
>
> The reviewer is right about us having stronger assumptions but:
>  - In the context of QRE solving these are assumptions are met.
>  - Other methods do iterate averaging so linear rates are likely not possible.
>  - Previous works do not take advantage of assumptions that hold in QRE problems.
>
> We elaborate on these points further below:
>
> When compared to other QRE algorithms for EFGs (Farina et al. (2019) and Ling et al. (2019)) it is true that we used different assumptions. However, this set of assumptions always holds for QRE problems. Indeed, part of the reason that we are able to attain a superior convergence guarantee is because we have identified special structure (as well as a new algorithm capable of exploiting said structure). In contrast, Ling et al. (2019) use an existing primal dual algorithm that does not take advantage of relative strong convexity (which holds for all QRE problems). Similarly, Farina et al. (2019) do not take advantage of the structure in QRE problems. Instead, Farina et al. (2019) use a no-regret learning reduction that is adversarial in nature and cannot accurately capture the dynamics of QRE solving. Moreover, it is unlikely that both the algorithms from Ling et al. (2019) and Farina et al. (2019) can attain linear convergence with modified assumptions since they use iterate averaging.

---

### Official Review · Reviewer_UyUG · 2022-10-24

**Confidence:** 2
**Correctness:** 3
**Technical Novelty And Significance:** 3
**Empirical Novelty And Significance:** 3
**Recommendation:** 8

**Clarity, Quality, Novelty And Reproducibility:**

The paper seems clear enough, the notation is extensive but this is to be expected. The results appear correct to me, I would prefer a comparison to more baselines in the experimental section though. The work seems to make a non-trivial contribution. The experiments are done on standard domains and the algorithm seems well described, so I believe the results to be replicable.

**Strength And Weaknesses:**

This is a highly technical work with an extensive appendix. The text reads relatively well, given the degree of technicality, and the authors seem to exert great effort to position their work in other literature on the topic. Still, I can't shake the feeling this paper would benefit from more carefully developed exposition, with more examples in the main text, and more detailed discussions on the introduced concepts. I understand this is not possible given the space limits.

In the short timeframe given by ICLR for reviewing I attempted to go through some of the theoretical results and I did not spot any obvious errors. I find Theorem 3.4 to be a nice result. Reiterating my previous concern regarding limited space dedicated to discussions though, I wonder how strong this theorem’s assumptions are in practice, especially in the context of reinforcement-learning problems. From the game-theoretic perspective, I also fail to find any other solution concept that would admit a VI formulation besides (logit) QREs. Are there some, for example, refinements of Nash equilibrium the authors know of that would be computable via MMD?

In addition, I remain confused about the convergence rates. The text mentions at multiple points the exponential convergence to QRE, yet the abstract and other parts claim linear convergence? What am I missing?

For completeness, I believe it would be also fair to discuss which Nash equilibria are actually reachable as limits of logit quantal response equilibria, i.e., to mention the convergence is pointwise to the best response if it is single valued, otherwise all actions in the best response are played with equal probabilities.

Finally, I also wonder why the authors did not compare their approach to other QRE-based algorithms like ADIDAS in normal-form games, since it is readily available in OpenSpiel? Why there is no comparison to other QRE algorithms in extensive-form games like the algorithms of Farina et al. or Ling et al. which this work refers to? Why is the comparison in the second experiment done with respect to CFR and not to (at least) CFR+? Also, how time consuming is one iteration of the MMD in contrast to other iterative algorithms mentioned in this work, i.e., how would the reported results look like with a temporal x-axis instead of plotting over iterations?


A few nitpicks:
The graphs in Fig 2 are missing y-axis labels.
Which quantity exactly is reported in Table 1? Are these utilities?


**Summary Of The Paper:**

This paper introduces an iterative optimization algorithm based on mirror descent that demonstrates competitive performance in reinforcement-learning tasks while provably converging to (logit) quantal response equilibrium in both normal-form and extensive-form two-player zero-sum games. The key idea of this algorithm - called Magnetic Mirror Descent (MMD) - appears to be to model these scenarios as variational inequality problems with monotone G operators that admit efficient optimization. The authors generalize a non-Euclidean proximal gradient method for this purpose. After formally proving the convergence in Theorem 3.4, the authors present a series of experimental results to substantiate their claims. First, they evaluate the convergence to quantal response equilibrium against Predictive Update and Optimistic Multiplicative Weight Update methods. In the second experiment they leverage the logit correspondence to reach Nash equilibrium and compare the speed of convergence over iterations to CFR. The last two experiments consider reinforcement-learning, either in single-agent setting, against PPO,  or in multi-agent setting, against NFSP.


**Summary Of The Review:**

This appears to be a theoretically well grounded paper that presents a new algorithm for solving variational inequality problems. It reads well and the theoretical results seem correct to me. I would appreciate if the empirical section considered more baselines to better evaluate the algorithm's performance though.

---

> ### Author Response · Authors · 2022-11-11
> **Author Response**
>
> Thanks for your thoughtful review! We agree with the reviewer’s characterization of the strengths and weaknesses of our work.
>
> > Still, I can't shake the feeling this paper would benefit from more carefully developed exposition.
>
> While we tried very hard to make it as readable as possible, we acknowledge that the submission as written is quite dense. We have incorporated suggestions of reviewers regarding the exposition into an updated draft in an attempt to make Section 3 more readable. Please let us know if you have any ideas on specific further improvements.
>
> > I find Theorem 3.4 to be a nice result.
>
> Thank you for liking our result! We went through great lengths to make our analysis readable and hope that our new perspective to QREs proves useful to others in the field.
>
> > Reiterating my previous concern regarding limited space dedicated to discussions though, I wonder how strong this theorem’s assumptions are in practice, especially in the context of reinforcement-learning problems.
>
> In the context of QRE solving, all of our theorem's assumptions are met. In the context of reinforcement learning, our theory does not explain the strong performance of MMD. We feel that theoretically characterizing MMD's performance as a reinforcement learning algorithm is an important open problem.
>
> > From the game-theoretic perspective, I also fail to find any other solution concept that would admit a VI formulation besides (logit) QREs. Are there some, for example, refinements of Nash equilibrium the authors know of that would be computable via MMD?
>
> Variational inequalities provides a framework to model many problems including Nash equilibria in general-sum games and any fixed point problem. However, given the class of variational inequalities we study (i.e. $\operatorname{VI}(\mathcal{Z}, F+\nabla g)$) we can apply our method to min max problems as defined by equation (2). For example, in Section D.5 we consider a Euclidean case. Additionally, our algorithm can also be used to approximately solve
> $$
>  \min_{x\in \mathcal{X}} \max_{y\in \mathcal{Y}} f(x,y)
> $$
> since any $\epsilon$-Nash equilibrium of the above game can be reached by setting $\alpha$ small enough in equation (2).
>
> Additionally, we add that $\operatorname{VI}(\mathcal{Z}, F+\nabla g)$ can model QREs in n-player general sum games; however, $F$ is no longer monotone in the general-sum case. We think that our work could be a stepping stone to study this general-sum QRE case and hope it encourages the algorithmic game theory community to consider this optimization/variational inequality perspective.
>
> > In addition, I remain confused about the convergence rates. The text mentions at multiple points the exponential convergence to QRE, yet the abstract and other parts claim linear convergence? What am I missing?
>
> Perhaps confusingly, exponential convergence and linear convergence are synonymous terms. It is called exponential because it converges exponentially fast and linear because the error is a linear function of the previous error. We have also included an external source for reference: https://www.cs.ubc.ca/~schmidtm/Courses/540-W18/L5.pdf (page 27).
>
> > For completeness, I believe it would be also fair to discuss which Nash equilibria are actually reachable as limits of logit quantal response equilibria, i.e., to mention the convergence is pointwise to the best response if it is single valued, otherwise all actions in the best response are played with equal probabilities.
>
> It is a known result that the limit of an $\alpha$-QRE as $\alpha \to 0$ is a Nash equilibrium. However, not every Nash equilibrium is the limit of an $\alpha$-QRE. For example, consider the 2p0s matrix game with payoff of all zeros. In this case, every joint policy is a Nash equilibrium but only the uniform random policy is a QRE.
>
> We kindly ask the reviewer if we have correctly understood their comment.
>
>
> > Finally, I also wonder why the authors did not compare their approach to other QRE-based algorithms like ADIDAS in normal-form games, since it is readily available in OpenSpiel? Why there is no comparison to other QRE algorithms in extensive-form games like the algorithms of Farina et al. or Ling et al. which this work refers to?
>
> The reason we did have these baselines is because, at the time of submission, we were not aware of ADIDAS and neither Farina et al. nor Ling et al. had open sourced code for their approaches.
>
> > Why is the comparison in the second experiment done with respect to CFR and not to (at least) CFR+?
>
> We have added CFR+ as a baseline for Figure 2.
>
> > The graphs in Fig 2 are missing y-axis labels.
>
> We specify in the caption that the plots are measured in exploitability. We have modified the plot to add this to the y-axis, also.
>
> > Which quantity exactly is reported in Table 1? Are these utilities?
>
> Thanks for catching this. These are expected returns -- we have updated the table caption.

---

### Official Review · Reviewer_Mj5Y · 2022-10-25

**Confidence:** 2
**Correctness:** 4
**Technical Novelty And Significance:** 4
**Empirical Novelty And Significance:** 3
**Recommendation:** 8

**Clarity, Quality, Novelty And Reproducibility:**

To the best of my knowledge the approach is novel, and appears to be reproducible.

**Strength And Weaknesses:**

Strengths:
* Provides the first algorithm to get empirically strong performance in both single-agent and multi-agent RL settings
* The theoretical convergence guarantees of magnetic mirror decent is the first time such guarantees have been shown in 2 player zero sum games with a first order solver.

Weaknesses:
* The biggest weakness of this paper is the clarity.  Given the subject matter, this is not the fault of the authors, but the authors themselves seem to be aiming for a broader audience than this paper is accessible to:
 ``
 We hope that, due to its simplicity, MMD will help open the door to 2p0s games research for RL researchers without game theoretic backgrounds.
 ''
I believe, how the paper is written now, such researchers would have find it difficult to understand the core method or contribution.  I think this is a challenging goal to set, but I think the authors are right to aim for it.  Given that this paper has the potential to be seminal, it could be the starting point for many RL researchers aiming to understand this subfield, it would be best if the paper offered enough direction for those researchers to be successful in understanding it.

I appreciate that the authors have already put some effort into doing this, it is claimed that Section 2.2 - Section 3.1 can be skipped.  However, there are a few points this misses:
* variables used in 3.2 are not defined if Section 2.2-3.1 are skipped (eta, zeta, psi)
* The term "magnet policy" is never explained even informally
* The core reason why the intuition as to why this approach works where other approaches have failed is not clear

There are also bits of prior knowledge that are assumed from readers:
* the term ``behavioral strategies'' is used but not explained
* It's not initially clear from context that the Gambit engine on page 6 is only used to measure performance of the algorithm, rather than as a part of the algorithm itself.
* The term "regularized equilibria" is used without explanation

Finally, within Section 2.2-3.1, it would be good to be accommodating to those readers as well, at least in having a short summary that outlines the main sections of the argument.  Ideally this would highlight what the key change is that other methods missed.  For instance, just flagging that you will cast it to a variational inference problem, show a dynamical system which converges to the solution of that variational inference problem, and then noticing that that dynamical system factors into separate updates.

**Summary Of The Paper:**

This paper presents a method, Magnetic Mirror Decent, which can get comparable performance to state of the art in both single-agent and multi-agent RL settings.  To the best of my knowledge it is the first algorithm known to have such strong empirical performance in both settings, and to have strong theoretical convergence guarantees in both settings.

**Summary Of The Review:**

This paper has the potential to be very impactful to the field, as such I am recommending acceptance.  However, it is quite dense, to some extent unavoidably.  I believe this can be mitigated to some extent, to allow the central result to be accessible to more readers.

---

> ### Author Response · Authors · 2022-11-11
> **Author Response**
>
> Thanks for your thoughtful review! We agree with the reviewer's characterization of the strengths and weaknesses of our work.
>
> > variables used in 3.2 are not defined if Section 2.2-3.1 are skipped (eta, zeta, psi)
> > The term "magnet policy" is never explained even informally
>
> Thanks for this feedback! We have updated the text to make sure all variables are re-defined after the end of the technical section.
>
> > The core reason why the intuition as to why this approach works where other approaches have failed is not clear
>
> For QRE solving, we would not necessarily characterize existing approaches as having failed. Rather, we would say they have attained weaker convergence guarantees. Unique to our approach is both our problem setup and analysis (variational inequalities). No previous works formulate the QRE problem as a variational inequality problem with composite structure like we have done in Proposition 2.4.
>
> For reinforcement learning, the reasons behind MMD's success remain unclear. We believe this to be an important theoretical open problem for multi-agent learning.
>
> > the term ``behavioral strategies'' is used but not explained
>
> We were not able to find anywhere in the text that we used the term "behavioral strategies". We do see that we used the term "behavioral-form algorithm". We tried to define it in the sentence in which it was first introduced: "MMD can also be a considered as a behavioral-form algorithm in which update rule (10) or (11) is applied at each information state." Please let us know if the reviewer was referring to a different usage that they feel is poorly explained.
>
> > It's not initially clear from context that the Gambit engine on page 6 is only used to measure performance of the algorithm, rather than as a part of the algorithm itself.
>
> Thanks for this feedback! We added a sentence to the "Convergence to Quantal Response Equilibria" to clarify this point.
>
> > The term "regularized equilibria" is used without explanation
>
> Thanks for this feedback! We added a parenthetical to Section 3.2 to clarify this point.
>
> > Finally, within Section 2.2-3.1, it would be good to be accommodating to those readers as well, at least in having a short summary that outlines the main sections of the argument. Ideally this would highlight what the key change is that other methods missed. For instance, just flagging that you will cast it to a variational inference problem, show a dynamical system which converges to the solution of that variational inference problem, and then noticing that that dynamical system factors into separate updates.
>
> Thank you for this suggestion! We have added a paragraph at the beginning of Section 3 to outline our technical approach, as suggested.
>
> > It is quite dense, to some extent unavoidably.
>
> While we tried very hard to make the submission as readable as possible, we agree with the reviewer's assessment of its density. Please let us know if you have any other suggestions to improve its readability.

---

### Official Review · Reviewer_BKWB · 2022-10-29

**Confidence:** 3
**Correctness:** 4
**Technical Novelty And Significance:** 1
**Empirical Novelty And Significance:** 1
**Recommendation:** 3

**Clarity, Quality, Novelty And Reproducibility:**

Clarity: Fair. Some terminology lacks a formal definition.

Quality: Good.

Novelty: Somewhat.

Reproducibility: I believe it should be fine.


**Strength And Weaknesses:**

Strength:
 - The paper is generally well written, with complete theory and numerical experiment details.

Weaknesses:
 - My biggest concern is about the motivation of designing a single RL algorithm for both single-agent RL and 2p0s. Extending single-agent methods to 2p0s seems reasonable for me: many 2p0s methods are at least inspired by single-agent methods. But the other way does not look reasonable since if we know we are solving single-agent RL, why bother using 2p0s methods? I think the authors need more elaboration on this.
 - Another concern is the novelty of the proposed method. MMD in (10) seems to add a “BC” term $KL(\pi, \rho)$ beyond MD, which from my understanding helps stabilize the algorithm in the single-agent setting (most works from offline paper adopted a similar idea: TD3-BC, BCQ, SCORE, etc). In 2p0s, this term seems to drive the policies toward an equilibrium in a uniform sense. In terms of the proof, it follows a standard MD proof on the functional level (no parameterization) with some extra work on the “BC” term.
 - I also have concerns about the experiments. The experiments seem to be too simple. For example, in single-agent RL, they only consider very limited atari and mujoco games.


**Summary Of The Paper:**

The paper proposed MMD, a variant of MD with an extra "behavior cloning" term, to unify single-agent RL and Two-Player Zero-Sum Games. The method is shown to converge linear in a functional sense. Some numerical experiments are provided to justify the algorithm.

**Summary Of The Review:**

Overall I think the current revision lacks a strong motivation. Given other concerns mentioned before, I recommend reject.

---

> ### Author Response · Authors · 2022-11-11
> **Author Response**
>
> Thanks for your thoughtful review!
>
> > Extending single-agent methods to 2p0s seems reasonable for me: many 2p0s methods are at least inspired by single-agent methods. But the other way does not look reasonable since if we know we are solving single-agent RL, why bother using 2p0s methods?
>
> We agree with the reviewer's characterization here. If a practitioner is faced with a single-agent learning problem, there is no reason to necessarily look at 2p0s methods. We would characterize the attractiveness of unification as going in the opposite direction -- i.e., our contribution allows single-agent reinforcement learning practitioners to approach 2p0s problems without having to understand 2p0s-specific literature.
>
> > My biggest concern is about the motivation of designing a single RL algorithm for both single-agent RL and 2p0s.
>
> While this is something that our submission does, our contribution is much more than this. Our approach breaks new ground for three different communities:
> 1. For variational inequalities, we provide the first non-Euclidean proximal gradient method to attain linear convergence.
> 2. For QRE solving, our approach is the first first-order method to achieve linear convergence in extensive-form games.
> 3. For reinforcement learning in 2p0s games, we give the first reinforcement learning algorithm to achieve competitive results with CFR.
>
> We respectfully point out that the reviewer seems to have missed this novelty in their assessments of *technical novelty and significance* and *empirical novelty and significance*, both of which the reviewer rated as "neither novel nor significant".
>
> > Another concern is the novelty of the proposed method. MMD in (10) seems to add a “BC” term  beyond MD, which from my understanding helps stabilize the algorithm in the single-agent setting (most works from offline paper adopted a similar idea: TD3-BC, BCQ, SCORE, etc).
>
> This is not exactly right. The purpose of the additional regularization term (in the context of our analysis) is to stabilize learning dynamics in minimax settings. The additional term need not be related to behavior cloning.
>
> We readily acknowledge that there are multiple existing works that investigate methods closely related to MMD, as we discussed in the related work section. However, although the works referenced by the reviewer also use regularization toward a reference policy, the algorithms remain different and are not studied in the context of two-player zero-sum games. Furthermore, (10) is only one instantiation of MMD -- our results are much more general than (10). For example, see equation (11) or Section D5 for examples of Euclidean MMD (i.e., $\psi = \frac{1}{2}\lVert \cdot \rVert^2$).
>
> > In terms of the proof, it follows a standard MD proof on the functional level (no parameterization) with some extra work on the “BC” term.
>
> While the proof itself uses standard techniques from both the variational inequality and mirror descent literature, the manner in which we combine them is not standard. For example, using the assumption of relative strong convexity from convex optimization within smooth monotone variational inequalities is novel. We respectfully emphasize to the reviewer that *Theorem 3.4 is the only known linear convergence result for a non-Euclidean proximal gradient method for smooth monotone variational inequality problems*. In contrast, standard mirror descent results generally differ for several reasons:
> 1. Mirror descent does not typically include a closed-form proximal term.
> 2. Results for variational inequalities are often given in the average iterate and do not show convergence of the divergence to the solution $B_\psi(z_\ast;z_t)$ (e.g., Nemirovski's Mirror-Prox method).
>
>
> > I also have concerns about the experiments. The experiments seem to be too simple. For example, in single-agent RL, they only consider very limited atari and mujoco games.
>
> The single-agent RL experiments are not primary contributions; they are more like sanity checks. Our most important empirical contributions are confirming linear QRE solving for EFGs and showing that MMD (as an RL algorithm) yields competitive results with CFR. Our secondary experimental contributions include extensively benchmarking MMD as a tabular equilibrium solver (see list under the *Subject Matter of Additional Experiments* header) and showing that MMD as a deep MARL algorithm can outperform NFSP.

---

### Decision · Program_Chairs · 2023-01-20

**Decision:**

Accept: poster

**Justification For Why Not Higher Score:**

The paper has relevant strong points but also some weaknesses, especially in terms of motivation and clarity of the presentation.

**Justification For Why Not Lower Score:**

The paper is borderline, but it presents some relevant contributions that deserve publication.

**Metareview: Summary, Strengths And Weaknesses:**

The paper proposes a reinforcement learning algorithm based on mirror descent, which can get good performance both in single-agent and multi-agent problems. The authors provide both theoretical and empirical analysis of the proposed work.
After reading each others' reviews and the authors' feedback, the reviewers discussed their concerns without reaching a consensus.
The main criticisms are related to the motivation, the novelty and the clarity of the presentation.
I think the authors have provided good replies to the first two points, while the third can be addressed in preparing the final version.
On the other hand, the strong points (technical strong, well-grounded, broad experimental analysis) represent relevant contributions for which the paper deserves to be accepted.
In preparing the final version of their paper, the authors need to pay particular attention to the clarity issues raised by the reviewers.

**Note From Pc:**

if the above contains the word "oral" or "spotlight" please see: "oral" presentation means -> notable-top-5% and "spotlight" means -> notable-top-25%. As stated in our emails, we are disassociating presentation type from AC recommendations

**Summary Of Ac-Reviewer Meeting:**

Only Reviewer Mj5Y and Reviewer UyUG participated in the virtual meeting.
They were the two reviewers in favour of the acceptance, and they confirmed their evaluation even in light of the criticisms expressed by the other two reviewers, who did not participate in the discussion either online or offline.